# Overfitting for Fun and Profit: Instance-Adaptive Data Compression

**Ties van Rozendaal** [*]
Qualcomm AI Research [†]
ties@qti.qualcomm.com

**Taco S. Cohen**
Qualcomm AI Research [†]
tacos@qti.qualcomm.com

**Iris A.M. Huijben**[*‡]
Qualcomm AI Research [†]

Department of Electrical Engineering
Eindhoven University of Technology
i.a.m.huijben@tue.nl

## Abstract

Neural data compression has been shown to outperform classical methods in terms of rate-distortion ($RD$) performance, with results still improving rapidly. At a high level, neural compression is based on an autoencoder that tries to reconstruct the input instance from a (quantized) latent representation, coupled with a prior that is used to losslessly compress these latents. Due to limitations on model capacity and imperfect optimization and generalization, such models will suboptimally compress test data in general. However, one of the great strengths of learned compression is that if the test-time data distribution is known and relatively low-entropy (e.g. a camera watching a static scene, a dash cam in an autonomous car, etc.), the model can easily be finetuned or *adapted* to this distribution, leading to improved $RD$ performance. In this paper we take this concept to the extreme, adapting the *full* model to a *single* video, and sending model updates (quantized and compressed using a parameter-space prior) along with the latent representation. Unlike previous work, we finetune not only the encoder/latents but the entire model, and - during finetuning - take into account both the effect of model quantization and the additional costs incurred by sending the model updates. We evaluate an image compression model on I-frames (sampled at 2 fps) from videos of the Xiph dataset, and demonstrate that full-model adaptation improves $RD$ performance by $\sim 1$ dB, with respect to encoder-only finetuning.

## 1 Introduction

The most common approach to neural lossy compression is to train a variational autoencoder (VAE)-like model on a training dataset to minimize the *expected RD* cost $D + \beta R$ (Theis et al., 2017; Kingma & Welling, 2013). Although this approach has proven to be very successful (Ballé et al., 2018), a model trained to minimize expected $RD$ cost over a full dataset is unlikely to be optimal for every test instance because the model has limited capacity, and both optimization and generalization will be imperfect. The problem of generalization will be especially significant when the testing distribution is different from the training distribution, as is likely to be the case in practice.

Suboptimality of the encoder has been studied extensively under the term *inference suboptimality* (Cremer et al., 2018), and it has been shown that finetuning the encoder or latents for a particular instance can lead to improved compression performance (Lu et al., 2020; Campos et al., 2019; Yang et al., 2020b; Guo et al., 2020). This approach is appealing as no additional information needs to be added to the bitstream, and nothing changes on the receiver side. Performance gains however are limited, because the prior and decoder can not be adapted.

---

[*]Equal contribution
[†]Qualcomm AI Research is an initiative of Qualcomm Technologies, Inc.
[‡]Work done during internship at Qualcomm AI Research

In this paper we present a method for *full-model instance-adaptive compression*, i.e. adapting the *entire* model to a *single* data instance. Unlike previous work, our method takes into account the costs for sending not only the latent prior, but also the decoder model updates, as well as quantization of these updates. This is achieved by extending the typical $RD$ loss with an additional model rate term $M$ that measures the number of bits required to send the model updates under a newly introduced *model prior*, resulting in a combined $RDM$ loss.

As an initial proof of concept, we show that this approach can lead to very substantial gains in $RD$ performance ($\sim 1$ dB PSNR gain at the same bitrate) on the problem of I-frame video coding, where a set of key frames, sampled from a video at 2 fps, are independently coded using an I-frame (image compression) model. Additionally, we show how the model rate bits are distributed across the model, and (by means of an ablation study) quantify the individual gains achieved by including a model-rate loss and using quantization-aware finetuning.

The rest of this paper is structured as follows. Section 2 discusses the basics of neural compression and related work on adaptive compression. Section 3 presents our method, including details on the $RDM$ loss, the choice of the model prior, its quantization, and the (de)coding procedure. In Sections 4 and 5 we present our experiments and results, followed by a discussion in Section 6.

## 2 PRELIMINARIES AND RELATED WORK

### 2.1 NEURAL DATA COMPRESSION

The standard approach to neural compression can be understood as a particular kind of VAE (Kingma & Welling, 2013). In the compression literature the encoder $q_\phi(\boldsymbol{z}|\boldsymbol{x})$ is typically defined by a neural network parameterized by $\phi$, with either deterministic output (so $q_\phi(\boldsymbol{z}|\boldsymbol{x})$ is one-hot) (Habibian et al., 2019) or with fixed uniform $[0, 1]$ noise on the outputs (Ballé et al., 2018). In both cases, sampling $z \sim q_\phi(\boldsymbol{z}|\boldsymbol{x})$ is used during training while quantization is used at test time.

The latent $\boldsymbol{z}$ is encoded to the bitstream using entropy coding in conjunction with a latent prior $p_\theta(\boldsymbol{z})$, so that coding $\boldsymbol{z}$ takes about $-\log p_\theta(\boldsymbol{z})$ bits (up to discretization). On the receiving side, the entropy decoder is used with the same prior $p_\theta(\boldsymbol{z})$ to decode $\boldsymbol{z}$ and then reconstruct $\hat{\boldsymbol{x}}$ using the decoder network $p_\theta(\boldsymbol{x}|\boldsymbol{z})$ (note that we use the same symbol $\theta$ to denote the parameters of the prior and decoder jointly, as in our method both will have to be coded and added to the bitstream).

From these considerations it is clear that the rate $R$ and distortion $D$ can be measured by the two terms in the following loss:

$$\mathcal{L}_{\text{RD}}(\phi, \theta) = \beta \underbrace{\mathbb{E}_{q_\phi(\boldsymbol{z}|\boldsymbol{x})}\left[-\log p_\theta(\boldsymbol{z})\right]}_{R} + \underbrace{\mathbb{E}_{q_\phi(\boldsymbol{z}|\boldsymbol{x})}\left[-\log p_\theta(\boldsymbol{x}|\boldsymbol{z})\right]}_{D}. \tag{1}$$

This loss is equal (up to the tradeoff parameter $\beta$ and an additive constant) to the standard negative evidence lower bound (ELBO) used in VAE training. The rate term of ELBO is written as a KL divergence between encoder and prior, but since $D_{\text{KL}}(q, p) = R - H[q]$, and the encoder entropy $H[q]$ is constant in our case, minimizing the KL loss is equivalent to minimizing the rate loss.

Neural video compression is typically decomposed into the problem of independently compressing a set of key frames (i.e. I-frames) and conditionally compressing the remaining frames (Lu et al., 2019; Liu et al., 2020; Wu et al., 2018; Djelouah et al., 2019; Yang et al., 2020a). In this work, we specifically focus on improving I-frame compression.

### 2.2 ADAPTIVE COMPRESSION

A compression model is trained on a dataset $\mathcal{D}$ with the aim of achieving optimal $RD$ performance on test data. However, because of limited model capacity, optimization difficulties, or insufficient data (resulting in poor generalization), the model will in general not achieve this goal. When the test data distribution differs from that of the training data, generalization will not be guaranteed even in the limit of infinite data and model capacity, and perfect optimization.

A convenient feature of neural compression however is that a model can easily be finetuned on new data or data from a specific domain. A model can for instance (further) be trained after deployment,

and Habibian et al. (2019) showed improved $RD$ gains after finetuning a video compression model on footage from a dash cam, an approach dubbed *adaptive compression* (Habibian et al., 2019).

In adaptive compression, decoding requires access to the adapted prior and decoder models. These models (or their delta relative to a pretrained shared model) thus need to be signaled. When the amount of data coded with the adapted model is large, the cost of signaling the model update will be negligible as it is amortized. However, a tradeoff exists, the more restricted the domain of adaptation, the more we can expect to gain from adaptation (e.g. an image compared to a video or collection of videos). In this paper we consider the case where the domain of adaptation is a set of I-frames from a single video, resulting in costs for sending model updates which become very relevant.

## 2.3 CLOSING THE AMORTIZATION GAP

Coding model updates can easily become prohibitively expensive when the model is adapted for every instance. However, if we only adapt the encoder or latents, no model update needs to be added to the bitstream, since the encoder is not needed for decoding as the latents are sent anyway. We can thus close, or at least reduce, the *amortization gap* (the difference between $q_\phi(\boldsymbol{z}|\boldsymbol{x})$ and the optimal encoder; Cremer et al. (2018)) without paying any bits for model updates. Various authors have investigated this approach: Aytekin et al. (2018); Lu et al. (2020) adapt the encoder, while Campos et al. (2019); Yang et al. (2020b); Guo et al. (2020) adapt the latents directly. This simple approach was shown to provide a modest boost in $RD$ performance.

## 2.4 ENCODING MODEL UPDATES

As mentioned, when adapting (parts of) the decoder or prior to an instance, model updates have to be added to the bitstream in order to enable decoding. Recent works have proposed ways to finetune parts of the model, while keeping the resulting bitrate overhead small. For instance Klopp et al. (2020) train a reconstruction error predicting network at encoding time, quantize its parameters, and add them to the bitstream. Similarly (Lam et al., 2019; 2020) propose to finetune all parameters or only the convolutional biases, respectively, of an artifact removal filter that operates after decoding. A sparsity-enforcing and magnitude-suppressing penalty is leveraged, and additional thresholding is applied to even more strictly enforce sparsity. The update vector is thereafter quantized using k-means clustering. Finally, Zou et al. (2020) finetune the latents in a meta-learning framework, in addition to updating the decoder convolutional biases, which are quantized by k-means and thereafter transmitted. All these methods perform quantization post-training, leading to a potentially unbounded reduction in performance. Also, albeit the use of regularizing loss terms, no valid proxy for the actual cost of sending model updates is adopted. Finally, none of these methods performs adaptation of the full model.

The field of model compression is related to our work as the main question to be answered is how to most efficiently compress a neural network without compromising on downstream task performance (Han et al., 2016; Kuzmin et al., 2019). Bayesian compression is closely related, where the model weights are sent under a model prior (Louizos et al., 2017; Havasi et al., 2018) as is the case in our method. Instead of modeling uncertainty in parameters, we however assume a deterministic posterior (i.e. point estimate). Another key difference with these works is that we send the model parameter updates relative to an existing baseline model, which enables extreme compression rates (0.02-0.2 bits/param). This concept of compressing updates has been used before in the context of federated learning (McMahan et al., 2017; Alistarh et al., 2017) as well. We distinguish ourselves from that context, as there the model has to be compressed during every iteration, allowing for error corrections in later iterations. We only transmit the model updates once for every data instance that we finetune on.

## 3 FULL-MODEL INSTANCE-ADAPTIVE COMPRESSION

In this section we present full-model finetuning on one instance, while (during finetuning) taking into account both model quantization and the costs for sending model updates. The main idea is described in Section 3.1, after which Section 3.2 and 3.3 respectively provide details regarding the model prior and its quantization. The algorithm is described in Section 3.4.

### 3.1 FINETUNING AT INFERENCE TIME

Full-model instance-adaptive compression entails finetuning of a set of *global model* parameters $\{\phi_\mathcal{D}, \theta_\mathcal{D}\}$ (obtained by training on dataset $\mathcal{D}$) on a single instance $\boldsymbol{x}$. This results in updated parameters $\phi, \theta$, of which only $\theta$ has to be signaled in the bitstream. In practice we only learn the changes with respect to the global model, and encode the *model updates* $\delta = \theta - \theta_\mathcal{D}$ of the decoding model. In order to encode $\delta$, we introduce a continuous model prior $p(\delta)$ to regularize these updates, and use the quantized counterpart $p[\bar\delta]$ for entropy (de)coding them (more on quantization in Section 3.3).

The overhead for sending quantized model update $\bar\delta$ is given by model rate $\overline{M} = -\log p[\bar\delta]$, and can be approximated by its continuous counterpart $M = -\log p(\delta)$ (see Appendix A.1 for justification). Adding this term to the standard $RD$ loss using the same tradeoff parameter $\beta$, we obtain the instance-adaptive compression objective:

$$\mathcal{L}_{\text{RDM}}(\phi, \delta) = \mathcal{L}_{\text{RD}}(\phi, \theta_\mathcal{D} + \bar\delta) + \beta \underbrace{\left(-\log p(\delta)\right)}_{M}. \tag{2}$$

At inference time, this objective can be minimized directly to find the optimal model parameters for transmitting datapoint $\boldsymbol{x}$. It takes into account the additional costs for encoding model updates ($M$ term), and incorporates model quantization during finetuning ($\mathcal{L}_{\text{RD}}$ evaluated at $\bar\theta = \theta_\mathcal{D} + \bar\delta$).

### 3.2 MODEL PRIOR DESIGN

A plethora of options exist for designing model prior $p(\delta)$ as any probability density function (PDF) could be chosen, but a natural choice for modeling parameter update $\delta = \theta - \theta_\mathcal{D}$ is to leverage a Gaussian distribution, centered around the zero-update. Specifically, we can define the model prior on the updates as a multivariate zero-centered Gaussian with zero covariance, and a single shared (hyperparameter) $\sigma$, denoting the standard deviation: $p(\delta) = \mathcal{N}(\delta \,|\, \mathbf{0}, \sigma \boldsymbol{I})$.

Note that this is equivalent to modeling $\theta$ by $p(\theta) = \mathcal{N}(\theta \,|\, \theta_\mathcal{D}, \sigma \boldsymbol{I})$.

When entropy (de)coding the quantized updates $\bar\delta$ under $p[\bar\delta]$, we must realize that even the zero-update, i.e. $\bar\delta = \mathbf{0}$, is not for free. We define these initial static costs as $\overline{M}_0 = -\log p[\bar\delta = \mathbf{0}]$. Because the mode of the defined model prior is zero, these initial costs $\overline{M}_0$ equal the minimum costs. Minimization of eq. (2) thus ensures that – after overcoming these static costs – any extra bit spent on model updates will be accompanied by a commensurate improvement in $RD$ performance.

Since our method works best when signaling the zero-update is cheap, we want to increase the probability mass $p[\bar\delta = \mathbf{0}]$. We propose to generalize our earlier proposed model prior to a so-called spike-and-slab prior (Ročková & George, 2018), which drastically reduces the costs for this zero-update. More specifically, we redefine the PDF as a (weighted) sum of two Gaussians – a wide (slab) and a more narrow (spike) distribution:

$$p(\delta) = \frac{p_{\text{slab}}(\delta) + \alpha \, p_{\text{spike}}(\delta)}{1 + \alpha}, \quad \text{with}$$

$$p_{\text{slab}}(\delta) = \mathcal{N}(\delta|\mathbf{0}, \sigma\boldsymbol{I}) \quad \text{and} \quad p_{\text{spike}}(\delta) = \mathcal{N}(\delta|\mathbf{0}, \frac{t}{6}\boldsymbol{I}), \tag{3}$$

where $\alpha \in \mathbb{R}_{\geq 0}$ is a hyperparameter determining the height of the spiky Gaussian with respect to the wider slab, $t$ is the the bin width used for quantization (more details in Section 3.3), and $\sigma >> \frac{t}{6}$. By choosing the standard deviation of the spike to be $\frac{t}{6}$, the mass within six standard deviations (i.e. 99.7% of the total mass) is included in the central quantization bin after quantization. Note that the (slab-only) Gaussian prior is a special case of the spike-and-slab Gaussian prior in eq. (3), where $\alpha = 0$. As such, $p(\delta)$ refers to the spike-and-slab prior in the rest of this work. Appendix A.2 compares the continuous and discrete spike-and-slab prior and its gradients.

Adding the spike distribution, not only decreases $\overline{M}_0$, it also more heavily enforces sparsity on the updates via regularizing term $M$ in eq. (2). In fact, a high spike (i.e. large $\alpha$) can make the bits for signaling a zero-update so small (i.e. almost negligible), that the model effectively learns to make a binary choice; a parameter is either worth updating at the cost of some additional rate, or its not updated and the 'spent' bits are negligible.

## 3.3 QUANTIZATION

In order to quantize a scalar $\delta_i$ (denoted by $\delta$ in this section to avoid clutter), we use $N$ equispaced bins of width $t$, and we define the following quantization function:

$$\bar{\delta} = Q_t(\delta) = \text{clip}\left(\left\lceil \frac{\delta}{t} \right\rfloor \cdot t, \ \min = -\frac{(N-1)t}{2}, \ \max = \frac{(N-1)t}{2}\right). \tag{4}$$

As both rounding and clipping are non-differentiable, the gradient of $Q_t$ is approximated using the Straight-Through estimator (STE), proposed by Bengio et al. (2013). That is, we assume $\partial Q_t(\delta)/\partial \delta = 1$.

The bins are intervals $Q_t^{-1}(\bar{\delta}) = [\bar{\delta} - t/2, \bar{\delta} + t/2]$. We view $t$ as a hyperparameter, and define $N$ to be the smallest integer such that the region covered by bins (i.e. the interval $[-(N-1)t/2, (N-1)t/2])$, covers at least $1 - 2^{-8}$ of the probability mass of $p(\delta)$. Indeed the number of bins $N$ is proportional to the ratio of the width of $p(\delta)$ and the width of the bins: $N \propto \sigma/t$. The number of bins presents a tradeoff between finetuning flexibility and model rate costs $\overline{M}$, so the $\sigma/t$-ratio is an important hyperparameter. The higher $N$, the higher these costs due to finer quantization, but simultaneously the lower the quantization gap, enabling more flexible finetuning.

Since $\bar{\delta} = Q_t(\delta)$, the discrete model prior $p[\bar{\delta}]$ is the pushforward of $p(\delta)$ through $Q_t$:

$$p[\bar{\delta}] = \int_{Q^{-1}(\bar{\delta})} p(\delta)d\delta = \int_{\bar{\delta}-t/2}^{\bar{\delta}+t/2} p(\delta)d\delta = P(\delta < \bar{\delta} + t/2) - P(\delta < \bar{\delta} - t/2). \tag{5}$$

That is, $p[\bar{\delta}]$ equals the mass of $p(\delta)$ in the bin of $\bar{\delta}$, which can be computed as the difference of the cumulative density function (CDF) of $p(\delta)$ evaluated at the edges of that bin.

## 3.4 ENTROPY CODING AND DECODING

After finetuning of the compression model on instance $\boldsymbol{x}$ by minimizing the loss in eq. (2), both the latents and the model updates are entropy coded (under their respective priors $p_{\bar{\theta}}(\boldsymbol{z})$ and $p[\bar{\delta}]$) into a bitstream $\boldsymbol{b}$. Decoding starts by decoding $\bar{\delta}$ using $p[\bar{\delta}]$, followed by decoding $\boldsymbol{z}$ using $p_{\bar{\theta}}(\boldsymbol{z})$ (where $\bar{\theta} = \bar{\delta} + \theta_{\mathcal{D}}$), and finally reconstructing $\hat{\boldsymbol{x}}$ using $p_{\bar{\theta}}(\boldsymbol{x} \mid \boldsymbol{z})$. The whole process is shown in Figure 1 and defined formally in Algorithms 1 and 2.

## 4 EXPERIMENTAL SETUP

### 4.1 DATASETS

The experiments in this paper use images and videos from the following two datasets:

**CLIC19** [1] The CLIC19 dataset contains a collection of natural high resolution images. It is conventionally divided into a professional set, and a set of mobile phone images. Here we merge the existing training folds of both sets and use the resulting dataset $\mathcal{D}$ to train our I-frame model. The corresponding validation folds are used to validate the global model performance.

**Xiph-5N 2fps** [2] The Xiph dataset contains a variety of videos of different formats. We select a representative sample of five videos ($5N$) from the set of 1080p videos (see Appendix B for details regarding selection of these samples). Each video is temporally subsampled to 2 fps to create a dataset of I-frames, referred to as Xiph-5N 2fps. Frames in all videos contain $1920 \times 1080$ pixels, and the set of I-frames after subsampling to 2 fps contain between 20 and 42 frames. The five selected videos are single-scene but multi-shot, and come from a variety of sources. Xiph-5N 2fps is used to validate our instance-adaptive data compression framework.

### 4.2 GLOBAL MODEL ARCHITECTURE AND TRAINING

Model rate can be restricted by (among others) choosing a low-complexity neural compression model, and amortizing the additional model update costs over a large number of pixels.

---

[1]`https://www.compression.cc/2019/challenge/`
[2]`https://media.xiph.org/video/derf/`

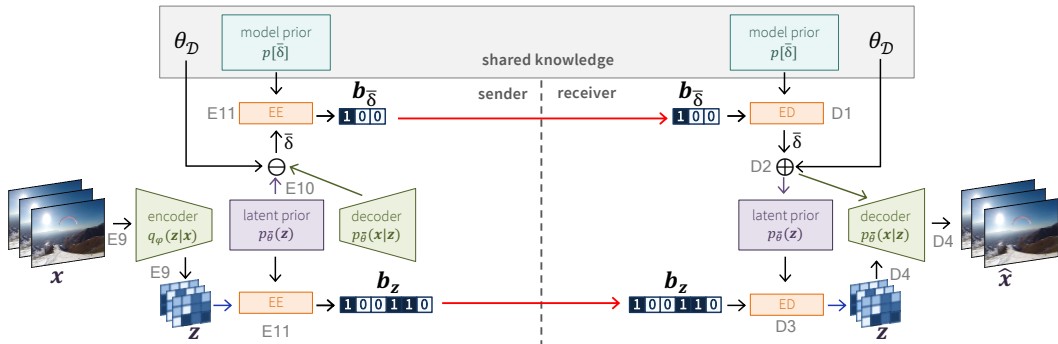

Figure 1: Visualization of encoding (Algorithm 1) and decoding (Algorithm 2) of our full-model instance-adaptive method. Each step is denoted with a code, e.g. *E9*, which refers to line 9 of the encoding algorithm. EE and ED denote entropy encoding and decoding, respectively. Both the latent representation $z$ and the parameter updates $\bar{\delta}$ are encoded in their respective bitstreams $b_z$ and $b_{\bar{\delta}}$. Model prior $p[\bar{\delta}]$ entropy decodes $b_{\bar{\delta}}$, after which the latent prior can decode $b_z$.

---

**Algorithm 1** Encoding of $x$

---

**Input:** Global model parameters $\{\phi_\mathcal{D}, \theta_\mathcal{D}\}$ trained on training set $\mathcal{D}$, model parameter quantizer $Q_t$, model prior $p[\delta]$, datapoint to be compressed $x$.

**Output:** Compressed bitstream $b = (b_{\bar{\delta}}, b_z)$

1: Initialize model parameters: $\phi = \phi_\mathcal{D}$, and $\theta = \theta_\mathcal{D}$
2: **for** step in MAX STEPS **do**
3:     Sample single I-frame: $x_f \sim x$
4:     Quantize transmittable parameters: $\bar{\theta} \leftarrow Q_t(\delta) + \theta_\mathcal{D}$, with $\delta = \theta - \theta_\mathcal{D}$
5:     Forward pass: $z \sim q_\phi(z \mid x_f)$ and evaluate $p_{\bar{\theta}}(x_f \mid z)$ and $p_{\bar{\theta}}(z)$.
6:     Compute loss $\mathcal{L}_{RDM}(\phi, \delta)$ on $x_f$ according to eq. (2).
7:     Backpropagate using STE for $Q_t$, then update $\phi, \theta$ using gradients $\frac{\partial \mathcal{L}_{RDM}}{\partial \phi}$ and $\frac{\partial \mathcal{L}_{RDM}}{\partial \theta}$.
8: **end for**
9: Compress $x$ to $z \sim q_\phi(x)$.
10: Compute quantized model parameters: $\bar{\theta} = \theta_\mathcal{D} + \bar{\delta}$, with $\bar{\delta} = Q_t(\theta - \theta_\mathcal{D})$.
11: Entropy encode: $b_{\bar{\delta}} = \text{enc}(\bar{\delta}; p[\bar{\delta}])$ and $b_z = \text{enc}(z; p_{\bar{\theta}}(z))$.

---

**Algorithm 2** Decoding of $x$

---

**Input:** Global model parameters $\theta_\mathcal{D}$ trained on training set $\mathcal{D}$, model prior $p[\bar{\delta}]$, bitstream $b = (b_{\bar{\delta}}, b_z)$.

**Output:** Decoded datapoint $\hat{x}$

1: Entropy decode: $\bar{\delta} = \text{dec}(b_{\bar{\delta}}; p[\bar{\delta}])$.
2: Compute updated parameters: $\bar{\theta} = \theta_\mathcal{D} + \bar{\delta}$.
3: Entropy decode latent under finetuned prior: $z = \text{dec}(b_z; p_{\bar{\theta}}(z))$.
4: Decode instance as the mean of the finetuned decoder: $p_{\bar{\theta}}(x|z)$

---

The most natural case for full-model adaptation is therefore to finetune a low-complexity model on a video instance. Typical video compression setups combine an image compression (I-frame) model to compress key frames, and a predict- or between-frame model to reconstruct the remaining frames. Without loss of generality for video-adaptive finetuning, we showcase our full-model finetuning framework for the I-frame compression problem, being a subproblem of video compression.

Specifically, we use the (relatively low-complexity) hyperprior-model proposed by Ballé et al. (2018), including the mean-scale prior (without context) from Minnen et al. (2018). Before finetuning, this model is trained on the training fold of the CLIC19 dataset, using the $RD$ objective given in eq. (1). Appendix C provides more details on both its architecture and the adopted training procedure.

### 4.3 INSTANCE-ADAPTIVE FINETUNING

Each instance in the Xiph-5N 2fps dataset (i.e. a set of I-frames belonging to one video) is (separately) used for full-model adaptation. The resulting model rate costs are amortized over the set of I-frames, rather than the full video. As a benchmark, we only finetune the encoding procedure, as it does not induce additional bits in the bitstream. Encoder-only finetuning can be implemented by either finetuning the encoding model parameters $\phi$, or directly optimizing the latents as in Campos et al. (2019). We implement both benchmarks, as the former is an ablated version of our proposed full-model finetuning, whereas the latter is expected to perform better due to the amortization gap Cremer et al. (2018).

Global models trained with $RD$ tradeoff parameter $\beta \in \{3e-3, 1e-3, 2.5e-4, 1e-4\}$ are used to initialize the models that are then being finetuned with the corresonding value for $\beta$. Both encoder-only and direct-latent finetuning minimize the $RD$ loss as given in eq. (1). For encoder-only tuning we use a constant learning rate of $1e-6$, whereas for latent optimization a learning rate of $5e-4$ is used for the low bitrate region (i.e. two highest $\beta$ values), and $1e-3$ for the high rate region. In case of direct latent optimization, the pre-quantized latents are finetuned, which are initialized using a forward pass of the encoder.

Our instance-adaptive *full-model* finetuning framework extends encoder-only finetuning by jointly updating $\phi$ and $\theta$ using the $RDM$ loss from eq. (2). In this case we finetune the global model that is trained with $\beta = 0.001$, independent of the value of $\beta$ during finetuning. Empirically, this resulted in negligible difference in performance compared to using the global model of the corresponding finetuning $\beta$, while it alleviated memory constraints thanks to the smaller size of this low bitrate model architecture (see Appendix C). The training objective in eq. (2) is expressed in bits per pixel and optimized using a fixed learning rate of $1e-4$. The parameters for the model prior were chosen as follows: quantization bin width $t = 0.005$, standard deviation $\sigma = 0.05$, and the multiplicative factor of the spike $\alpha = 1000$. We empirically found that sensitivity to changing $\alpha$ in the range 50-5000 was low. Realize that, instead of empirically setting $\alpha$, its value could also be solved for a target initial cost $\overline{M}_0$, given the number of decoding parameters and pixels in the finetuning instance.

All finetuning experiments (both encoding-only and *full-model*) ran for 100k steps, each containing one mini-batch of a single, full resolution I-frame[3]. We used the Adam optimizer (default settings) (Kingma & Ba, 2014), and best model selection was based on the $RD\overline{M}$ loss over the set of I-frames.

## 5 RESULTS

### 5.1 RATE-DISTORTION GAINS

Figure 2a shows the compression performance for encoder-only finetuning, direct latent optimization, and full-model finetuning, averaged over all videos in the Xiph-5N 2fps dataset for different rate-distortion tradeoffs. Finetuning of the entire parameter-space results in much higher $RD\overline{M}$ gains (on average approximately 1 dB for the same bitrate) compared to only finetuning the encoding parameters $\phi$ or the latents directly. Figure 8 in Appendix E shows this plot for each video separately. Note that encoder-only finetuning performance is on par with direct latent optimization, implying that the amortization gap (Cremer et al., 2018) is close to zero when finetuning the encoder model on a moderate number of I-frames.

Table 1 provides insight in the distribution of bits over latent rate $R$ and model rate $\overline{M}$, which both increase for lower values of $\beta$. However, the relative contribution of the model rate increases for the higher bitrate regime, which could be explained by the fact that in this regime the latent rate can more heavily be reduced by finetuning. Figure 2a indeed confirms that the the total rate reduction is higher for the high bitrate regime, which thus fully originates from the reduction in latent rate after finetuning. Table 1 also shows that the static intial costs $\overline{M}_0$ only marginally contribute to the total model rate. Figure 2b shows for one video how finetuning progressed over training steps, confirming that the compression gains already at the beginning of finetuning cover these initial costs. This effect was visible for all tested videos (see Appendix E).

---

[3]All frames in the videos in Xiph-5N 2fps are of spatial resolution $1920 \times 1080$. In order to make this shape compatible with the strided convolutions in our model, we pad each frame to $1920 \times 1088$ before encoding. After reconstructing $\hat{x}$, it is cropped back to its original resolution for evaluation.

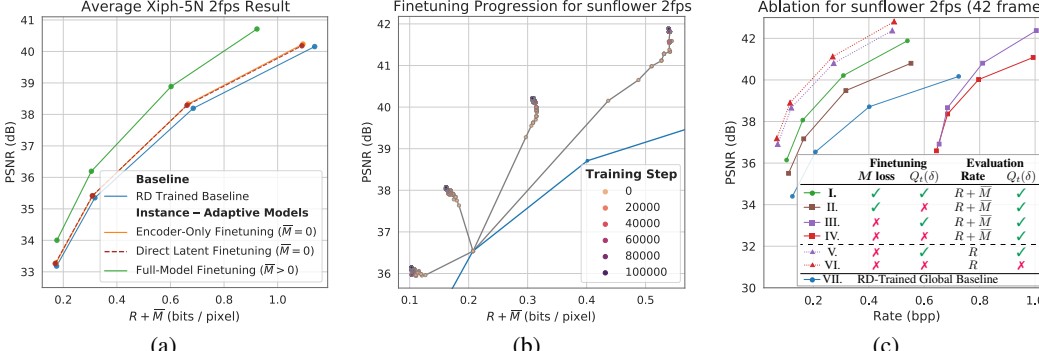

Figure 2: (a) Averaged $RD\overline{M}$ performance over all videos of the Xiph-5N 2fps datase for four different rate-distortion tradeoffs with $\beta = 3e{-}3$, $1e{-}3$, $2.5e{-}4$, and $1e{-}4$ (from left to right). Our *full-model* finetuning outperforms *encoder-only* and *direct latent* optimization with approximately 1 dB gain for the same rate. (b) Finetuning progression of the `sunflower` video over time. Between each dot, 500 training steps are taken, showing that already at the start of finetuning large $RD\overline{M}$ gains are achieved and the $RD\overline{M}$ performance continues to improve during finetuning. (c) Ablation where we show the effect of both quantization- ($Q_t(\delta)$) and model rate aware ($M$ Loss) finetuning. Case VI shows the upper bound on achievable finetuning performance when (naively) not taking into account quantization and model update rate.

## 5.2 ABLATIONS

Figure 2c shows several ablation results for one video. Case I is our proposed full-model finetuning, optimizing the $RDM$ loss, while simultaneously quantizing the updates to compute distortion (denoted with $Q_t(\delta)$). One can see that not doing quantization-aware finetuning (case II) deteriorates the distortion gains during evaluation. Removing the (continuous) model rate penalty from the finetuning procedure (case III) imposes an extreme increase in rate during evaluation, caused by unbounded growing of the model rate during finetuning. As such, the models result in a rate even much higher than the baseline model's rate, showing that finetuning without model rate awareness provides extremely poor results. Case IV shows performance deterioration in the situation of both quantization- and model rate unaware finetuning. Analyzing these runs while (naively) not taking into account the additional model update costs (cases V and VI) provides upper bounds on compression improvement. Case VI shows the most naive bound; $RD$ finetuning without quantization, whereas case V is a tighter bound that does include quantization. The gap between V and VI is small, suggesting that the used quantization strategy does only mildly harm performance.

An ablation study on the effect of the number of finetuning frames is provided in Appendix D. It reveals that, under the spike-and-slab model prior, full-model finetuning works well for a large range of instance lengths. Only in the worst case scenario, when finetuning only one frame in the low bitrate regime, full-model finetuning was found to be too costly.

## 5.3 DISTRIBUTION OF BITS

Figure 3 shows for different (mutually exclusive) parameter groups the distribution of the model updates $\delta$ (top) and their corresponding bits (bottom). Interestingly one can see how for (almost) all groups, the updates are being clipped by our earlier defined quantizer $Q_t$. This suggests the need for large, expensive updates in these parameter groups, for which the additional $RD$ gain thus appears to outweigh the extra costs. At the same time, all groups show an elevated center bin, thanks to training with the spike-and-slab prior (see Appendix F). By design of this prior, the bits paid for this zero-update are extremely low, which can best be seen in the bits histogram (Fig. 3-bottom) of the *Codec Decoder Weight* and *Biases*. The parameter updates of the *Codec Decoder IGND* group are the only ones that are non-symmetrically distributed across the zero-updated, which can be explained by the fact that IGDN (Ballé et al., 2016) is an (inverse) normalization layer. The *Codec Decoder Weights* were found to contribute most to the total model rate $\overline{M}$.

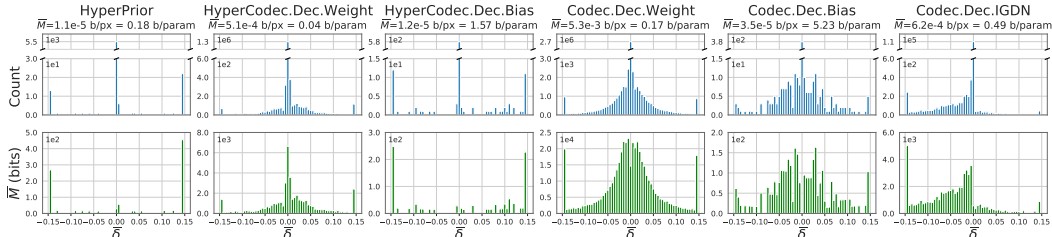

Figure 3: Empirical distribution of parameter updates for the model finetuned on the `sunflower` video with $\beta = 2.5e-4$. Columns denote parameter groups. Top: Histograms of the model updates $\bar{\delta}$. Bottom: Histogram of bit allocation for $\bar{\delta}$. Subtitles indicate the total number of bits for each parameter group, both expressed in bits per pixel (b/px) and bits per parameter (b/param).

| $\beta$ | PSNR (dB) | $R + \overline{M}$ | $R$ | $\overline{M}$ | $\overline{M}_0$ | $\overline{M}$ (bits/parameter) | $\overline{M}$ (kB/frame) |
|---|---|---|---|---|---|---|---|
| | | | (bits/pixel) | | | | |
| 3.0e-03 | 34.0 | 0.175 | 0.174 | 0.001 | 0.00033 | 0.026 | 0.39 |
| 1.0e-03 | 36.2 | 0.304 | 0.301 | 0.003 | 0.00033 | 0.044 | 0.67 |
| 2.5e-04 | 38.9 | 0.601 | 0.593 | 0.008 | 0.00033 | 0.129 | 1.99 |
| 1.0e-04 | 40.7 | 0.921 | 0.905 | 0.017 | 0.00033 | 0.282 | 4.35 |

Table 1: Distribution of bitrate for different rate-distortion tradeoffs $\beta$, averaged over the videos in the Xiph-5N 2fps dataset. The number of bits are distributed over the latent rate $R$ and the model rate $\overline{M}$, which is computed using the quantized model prior $p[\bar{\delta}]$.

## 6 DISCUSSION

This work presented instance-adaptive neural compression, the first method that enables finetuning of a full compression model on a set of I-frames from a single video, while restricting the additional bits for encoding the (quantized) model updates. To this end, the typical rate-distortion loss was extended by incorporating both model quantization and the additional model rate costs during fine-tuning. This loss guarantees pure $RD\overline{M}$ performance gains after overcoming a small initial cost for encoding the zero-update. We showed improved $RD\overline{M}$ performance on all five tested videos from the Xiph dataset, with an average distortion improvement of approximately 1 dB for the same bitrate.

Among videos, we found a difference in achieved finetuning $RD\overline{M}$ gain (see Appendix E). Possible causes can be three-fold. First, the performance of the global model differs per video, therewith influencing the maximum gains to be achieved by finetuning. Second, video characteristics such as (non-)stationarity greatly influence the diversity of the set of I-frames, thereby affecting the ease of model-adaption. Third, the number of I-frames differs per video and thus trades off model update costs (which are amortized over the set of I-frames), with ease of finetuning.

The results of the ablation in Fig. 2c show that the quantization gap (V vs VI) is considerably smaller than the performance deterioration due to (additionally) regularizing the finetuning using the model prior (I vs V). Most improvement in future work is thus expected to be gained by leveraging more flexible model priors, e.g. by learning its parameters and/or modeling dependencies among updates.

We showed how instance-adaptive full-model finetuning greatly improves $RD\overline{M}$ performance for I-frame compression, a subproblem in video compression. Equivalently, one can exploit full-model finetuning to enhance compression of the remaining (non-key) frames of a video, compressing the total video even further. Also, neural video compression models that exploit temporal redundancy, could be finetuned, as long as the model's complexity is low enough to restrict model rate. Leveraging such a low-complexity video model moves computational complexity of data compression from the receiver to the sender by heavily finetuning this small model on each video. We foresee convenience of this computational shift in applications where bandwidth and receiver compute power is scarce, but encoding compute time is less important. This use case in practice happens e.g. for (non-live) video streaming to low-power edge devices. Finding such low-complexity video compression models is however non-trivial as it's capacity must still be enough to compress an entire video. We foresee great opportunities here, and will therefore investigate this in future work.

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

# A  MODEL RATE LOSS

## A.1  GRADIENT DEFINITION

The proof below shows that $\nabla_\delta \overline{M}$ is an unbiased first-order approximation of $\nabla_\delta M$, validating its use during finetuning.

The gradient of the continuous model rate loss $M$ towards $\delta$ is defined as:

$$\nabla_\delta M = -\nabla_\delta \log p(\delta) = -\frac{\nabla_\delta p(\delta)}{p(\delta)} \tag{6}$$

The gradient of the non-differentiable discrete model rate loss $\bar{M}$ towards $\delta$ can be defined by exploiting the Straight-Through gradient estimator (Bengio et al., 2013), i.e. $\nabla_\delta \bar{\delta} = 1$.

As such[4]:

$$\nabla_\delta \overline{M} = -\nabla_\delta \log p[\bar{\delta}] = -\nabla_\delta \log [P(\delta < \bar{\delta} + \frac{t}{2}) - P(\delta < \bar{\delta} - \frac{t}{2})]$$

$$= -\frac{p(\bar{\delta} + \frac{t}{2}) - p(\bar{\delta} - \frac{t}{2})}{P(\delta < \bar{\delta} + \frac{t}{2}) - P(\delta < \bar{\delta} - \frac{t}{2})}. \tag{7}$$

By first-order approximation we can write:

$$\nabla_\delta p(\bar{\delta}) \approx \frac{p(\bar{\delta} + \frac{t}{2}) - p(\bar{\delta} - \frac{t}{2})}{t} \tag{8}$$

and

$$P(\delta < \bar{\delta} + \frac{t}{2}) - P(\delta < \bar{\delta} - \frac{t}{2}) \approx p(\bar{\delta})t. \tag{9}$$

Using eq. (8) and eq. (9), we can express the gradient of the discrete model rate costs $\nabla_\delta \bar{M}$ as:

$$\nabla_\delta \overline{M} \approx -\frac{\nabla_\delta p(\bar{\delta})t}{p(\bar{\delta})t} = -\frac{\nabla_\delta p(\bar{\delta})}{p(\bar{\delta})}, \tag{10}$$

which is thus a first-order approximation of $\nabla_\delta M$.

## A.2  CONTINUOUS VS DISCRETE MODEL RATE PENALTY

The proof provided in the previous section is not restricted to specific designs for $p(\delta)$, and thus holds both for the spike-and-slab prior including a spike ($\alpha > 0$), or the special case where no spike is used ($\alpha = 0$).

Figure 4 shows quantization of the model updates (top-left) and its corresponding gradient (left-bottom), exploiting the Straight-Through gradient estimator Bengio et al. (2013). Also the discrete (true) model rate $\bar{M}$ and its continuous analogy $M$ (middle-top) with their corresponding gradients (middle-bottom) are shown for the special case of not using a spike, i.e. $\alpha = 0$. One can see that the continuous model rate proxy is a shifted version of the discrete costs. Since such a translation does not influence the gradient (and neither gradient-based optimization), the continuous model rate loss can be used during finetuning, preventing instable training thanks to its smooth gradient.

---

[4]Due to cutting (a maximum) of $2^{-8}$ mass from the tails of $p(\delta)$ to enable quantization, the difference in cumulative masses in eq. (7) should be renormalized by $1 - 2^{-8}$. As this is a constant division inside a $\log$, it results in a subtraction of $\nabla_\delta \log 1 - 2^{-8} = 0$. Since this normalization does not influence the gradient, we omitted it for the sake of clarity.

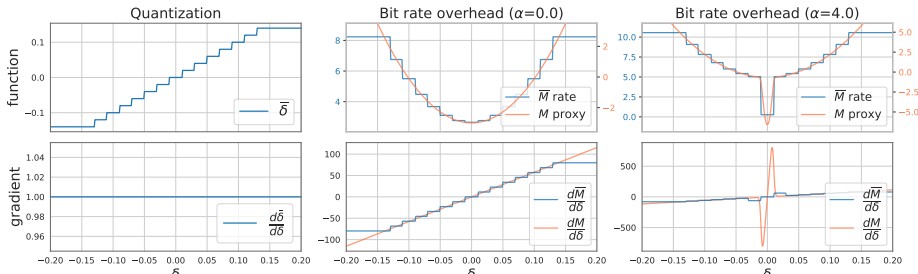

Figure 4: (Left) Illustrative example of the quantization effect and the corresponding gradient (using the Straight-Through estimator) for parameter update $\delta$. (Middle) The true bitrate overhead (blue) and its continuous proxy (orange) of a (slab-only) Gaussian prior ($\alpha = 0$) and their gradients with respect to unquantized $\delta$. (Right) The true bitrate overhead (blue) and its continuous proxy (orange) of a spike-and-slab prior ($\alpha = 4$) and their gradients with respect to unquantized $\delta$. One can see how the effect of the spike (almost) fully disappears in the gradient of the quantized bitrate overhead, as the largest amount of mass of the spike distribution is part of the center quantization bin.

Figure 4-right shows the same figure but for a model prior that includes a spike. Comparing true (discrete) model rate costs of the slab-only prior (middle-top), to these costs using the spike-and-slab prior (right-top), shows how the introduced spike reduces the number of bits to encode the zero-update (i.e. the center bin), at the cost of making larger updates more expensive in number of bits. Another interesting phenomena is visible when comparing the gradients of the discrete and continuous model rates for slab-only (middle-bottom) versus spike-and-slab prior (right-bottom). The effect of the spike almost fully disappears in the gradient of the discrete model rate. This is caused by the fact that most of the spike's mass is (by design) positioned inside the center quantization bin after quantization. Mathematically, this can be seen from filling in eq. (7) for the spike-and-slab prior:

$$\nabla_\delta \overline{M} = -\frac{p_{\text{slab}}(\bar{\delta}+\frac{t}{2}) - p_{\text{slab}}(\bar{\delta}-\frac{t}{2}) + \alpha\, p_{\text{spike}}(\bar{\delta}+\frac{t}{2}) - \alpha\, p_{\text{spike}}(\bar{\delta}-\frac{t}{2})}{P_{\text{slab}}(\delta < \bar{\delta}+\frac{t}{2}) - P_{\text{slab}}(\delta < \bar{\delta}-\frac{t}{2}) + \alpha\, P_{\text{spike}}(\delta < \bar{\delta}+\frac{t}{2}) - \alpha\, P_{\text{spike}}(\delta < \bar{\delta}-\frac{t}{2})}. \tag{11}$$

We can distinguish different behavior of eq. (11) for two distinct ranges of $\bar{\delta}$:

- $\bar{\delta} = 0$
  Symmetry around zero-update: $p_{\text{slab}}(\bar{\delta}+\frac{t}{2}) = p_{\text{slab}}(\bar{\delta}-\frac{t}{2})$ and $p_{\text{spike}}(\bar{\delta}+\frac{t}{2}) = p_{\text{spike}}(\bar{\delta}-\frac{t}{2})$.
  $\implies \nabla_\delta \overline{M} = 0$

- $\bar{\delta} \neq 0$
  Since $t = 6\sigma_{\text{spike}}$:
  $p_{\text{spike}}(\bar{\delta}+\frac{t}{2}) \approx 0, \quad p_{\text{spike}}(\bar{\delta}-\frac{t}{2}) \approx 0, \quad P_{\text{spike}}(\delta < \bar{\delta}+\frac{t}{2}) \approx 0, \quad P_{\text{spike}}(\delta < \bar{\delta}-\frac{t}{2}) \approx 0.$
  $\implies \nabla_\delta \overline{M} = -\frac{p_{\text{slab}}(\bar{\delta}+\frac{t}{2}) - p_{\text{slab}}(\bar{\delta}-\frac{t}{2})}{P_{\text{slab}}(\delta < \bar{\delta}+\frac{t}{2}) - P_{\text{slab}}(\delta < \bar{\delta}-\frac{t}{2})}.$
  Note that this approximation becomes less tight for large $\alpha$, as the small probabilities and cumulative densities of the spike distribution are multiplied by $\alpha$.

From the $\bar{\delta} = 0$ case, one can see that inclusion of the spike leaves the gradient unbiased. The $\bar{\delta} \neq 0$ case shows that the spike does not influence the gradient in the limit when the spike's mass is entirely positioned within the center quantization bin. As the standard deviation of the spiky Gaussian was chosen to be $\frac{t}{6}$, a total of $0.3\%$ of it's mass is in practice being quantized in an off-center quantization bin. This explains the slight increase of the off-center bins in the gradient of the discrete model rate costs in Fig. 4 (right-bottom). Comparing the gradient of the discrete versus continuous model rate costs for the spike-and-slab prior in Fig. 4 (right-bottom), we can see that the first order approximation between the two introduces a larger error than in the slab-only case (Fig. 4(middle-bottom)). This can be explained by the fact that the introduced spiky Gaussian has a higher tangent due to its support being more narrow than that of the Gaussian slab. Nevertheless, the use of the continuous model rate loss is preferred for finetuning as it (much more) strictly enforces zero-updates (thanks to the present gradient peaks around the center bin) than its discrete counterpart.

## B SAMPLE SELECTION FROM XIPH DATASET

### B.1 XIPH DATASET

The Xiph test videos can be found at `https://media.xiph.org/video/derf/`. Like Rippel et al. (2019) we select all 1080p videos, and exclude computer-generated videos and videos with inappropriate licences, which leaves us with the following videos:

```
aspen_1080p              ducks_take_off_1080p50   red_kayak_1080p          speed_bag_1080p
blue_sky_1080p25         in_to_tree_1080p50       riverbed_1080p25         station2_1080p25
controlled_burn_1080p    old_town_cross_1080p50   rush_field_cuts_1080p    sunflower_1080p25
crowd_run_1080p50        park_joy_1080p50         rush_hour_1080p25        tractor_1080p25
dinner_1080p30           pedestrian_area_1080p25  snow_mnt_1080p           west_wind_easy_1080p
```

### B.2 XIPH-5N 2 FPS DATASET

Due to computational limits, we draw a sample of five videos, which we refer to as the Xiph-5N dataset. When drawing such a small sample randomly, a high probability arises of drawing an unrepresentative sample, including for example too many videos with either low or high finetuning potential. To alleviate this problem, we use the following heuristic to select $N(= 5)$ videos:

1. Evaluate the global model's $RD$ performance on all videos for all values of $\beta$.
2. Per $\beta$ value, rank all videos based on their respective $\mathcal{L}_{RD}$ loss.
3. For each video, average the $\mathcal{L}_{RD}$ rank over all values of $\beta$.
4. Order all videos according to their average rank, and select $N$ videos based on $N + 1$ evenly spaced percentiles.

The global model's $RD$ performance for all videos from the Xiph dataset is shown in Figure 5. The five videos part of Xiph-5N are indicated with colors, and Table 2 provides more details about these five videos. The column titled *RD-tank percentile* shows the actual percentile at which the selected videos are ranked. The computed (target) percentiles for $N = 5$ are 1/6, 2/6, ..., 5/6. For each percentile we selected the video closest to these target percentiles. The last column denotes the number of I-frames after subsampling to 2 fps. Videos of which the original sampling frequency was not integer divisible by a factor 2, were subsampled with a factor resulting in the I-frame sampling frequency closest to 2 fps.

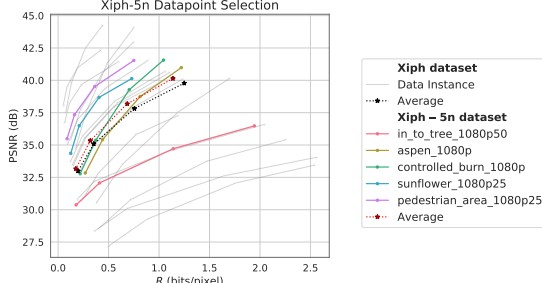

Figure 5: $RD$ performance of global baseline for all datapoints in Xiph and Xiph-5N.

| Video | RD-rank percentile | Target percentile | width × height × frames | Original fps | Duration (s) | Nr. of I-frames at 2 fps |
|---|---|---|---|---|---|---|
| `in_to_tree` | 0.22 | 0.167 | 1920 × 1080 × 500 | 50 | 10.0 | 20 |
| `aspen` | 0.37 | 0.333 | 1920 × 1080 × 570 | 30 | 19.0 | 38 |
| `controlled` | 0.50 | 0.500 | 1920 × 1080 × 570 | 30 | 19.0 | 38 |
| `sunflower` | 0.69 | 0.667 | 1920 × 1080 × 500 | 25 | 20.0 | 42 |
| `pedestrian_area` | 0.82 | 0.833 | 1920 × 1080 × 375 | 25 | 15.0 | 32 |
| **AVERAGE** | **0.52** | **0.500** | **1920 × 1080 × 503** | **32** | **16.6** | **34** |

Table 2: Characteristics of the five selected videos in Xiph-5N 2fps.

## C   GLOBAL MODEL ARCHITECTURE AND TRAINING

For our neural compression model, we adopt the architecture proposed by Ballé et al. (2018), including the mean-scale prior from Minnen et al. (2018). We use a shared hyperdecoder to predict the mean and scale parameters. Like (Ballé et al., 2018) we use a model architecture with fewer parameters for the low bitrate regime ($\beta \geq 1e-3$). Table 3 indicates both the model architecture and the number of parameters, grouped per sub-model. The upper row in this table links the terminology proposed by Ballé et al. (2018) to conventional VAE terminology which we follow in this work.

Figure 6 provides a visual overview of the model architecture, where we use $z_2$ and $z_1$ to indicate the latent and hyper-latent space respectively (referred to as $y$ and $z$ in the original paper of Ballé et al. (2018)). Note that even though we adopt a hierarchical latent variable model, we simplify the notation by defining a single latent space $z = \{z_1, z_2\}$ throughout this work.

During training we adopt a mixed quantization strategy, where the quantized latent $z_2$ are used to calculate the distortion loss (with their gradients estimated using the Straight-Trough estimator from Bengio et al. (2013)) , while we use noisy samples for $z_1, z_2$ when computing the rate loss.

Optimization of eq. (1) on the training fold of the CLIC19 dataset, was done using the Adam optimizer with default settings (Kingma & Ba, 2014), and took 1.5M steps for the low bitrate models, and 2.0M steps for the high bitrate models. Each step contained 8 random crops of $256 \times 256$ pixels, and the initial learning rate was set to $1e-4$ and lowered to $1e-5$ at 90% of training.

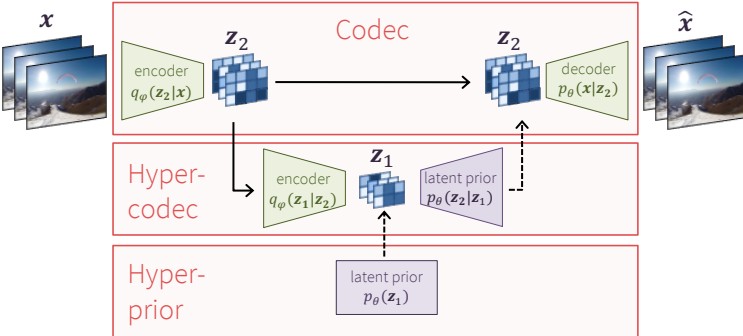

Figure 6: The mean-scale hyperprior model architecture visualized using the VAE framework.

Table 3: Model architecture and parameter count for the adopted hyperprior model. As suggested by Ballé et al. (2018), a distinct architecture is used for the low and high bitrate regime. We report the number of output channels per layer, for the exact model architecture we refer to Ballé et al. (2018). Note that the last column refers to the total number of parameters that needs to be known at the receiver side.

| | Transmitter | | Receiver | | | |
| --- | --- | --- | --- | --- | --- | --- |
| | Encoder $q_\phi(z_2\|x)$ | Hyper Encoder $q_\phi(z_1\|z_2)$ | Hyperprior $p_\theta(z_1)$ | Hyper Decoder $p_\theta(z_2\|z_1)$ | Decoder $p_\theta(x\|z_2)$ | Nr. of receiver parameters $\theta$ |
| **Low bitrate model** | | | | | | |
| Layers x Output channels | 4x192 | 3x128 | 3x3 | 2x128 + 1x256 | 3x192 + 1x3 | - |
| Parameter count | 2.89M | 1.04M | 5.50k | 1.26M | 2.89M | 4.16M |
| **High bitrate model** | | | | | | |
| Layers x Output channels | 4x320 | 3x192 | 3x3 | 2x192 + 1x384 | 3x320 + 1x3 | - |
| Parameter count | 8.01M | 2.40M | 8.26k | 2.95M | 8.01M | 10.97M |

# D    TEMPORAL ABLATION

In this experiment we investigate the tradeoff between the number of frames that the model is fine-tuned on, and the final $RD\overline{M}$ performance. The higher this number of frames, the higher (potentially) the diversity (making finetuning more difficult), but the lower the bitrate overhead (in bits/pixels) due to model updates. This ablation repeats our main experiment on the `sunflower` video (Fig. 8) for a varying number of I-frames.

We sample $f$ number of frames (equispaced) from the full video, starting at the zero'th index. The experiment is run for $f \in \{1, 2, 5, 10, 25, 50, 100, 250, 500\}$, and the two outmost rate-distortion tradeoffs: $\beta \in \{0.003, 0.0001\}$. Note that the original experiment was done with frames sampled at 2 fps, resulting in $f = 42$ for the `sunflower` video.

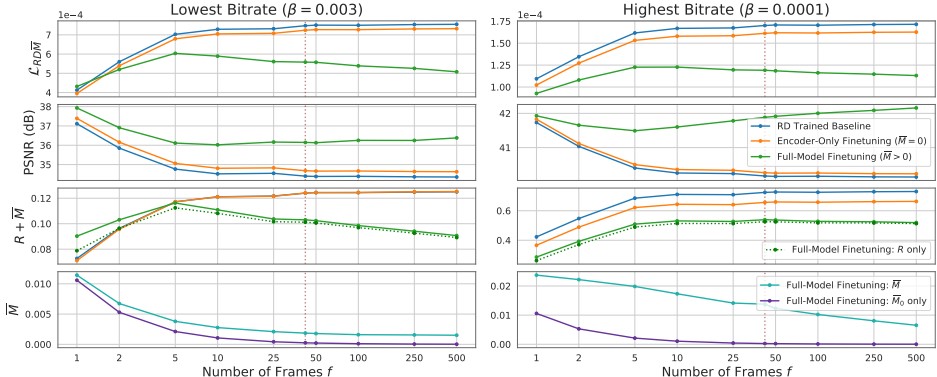

Figure 7: Compression performance as a function of number of finetuning frames for the `sunflower` video. The dashed red line indicates sampling at 2 fps (used in our main experiments).

Figure 7 shows (for the low and high bitrate region) the total $RD\overline{M}$ loss, and its subdivision in distortion and the different rate terms, as a function of numbers of finetuning frames. Full-model finetuning outperforms encoder-only finetuning in all cases, except for $f = 1$ in the low bit-rate regime. In this case, the model rate $\overline{M}$ is causing the total rate $R + \overline{M}$ to become too high to be competitive with the baselines. This in turn is mainly caused by the initial cost $\overline{M}_0$, which can only be amortized over a single frame. In general, for other values of $f$, these initial costs were found to contribute only little to the total rate (with even a negligible contribution in the low and high bitrate regions for respectively $f \geq 25$ and $f \geq 10$).

Note that the global model's performance varies noticeably as a function of $f$. Apparently, the first frame of this video is easy to compress, therewith lowering the total loss for small sets of I-frames. To make fair comparisons, one should thus only consider relative performance of encoder-only and full-model finetuning with respect to the $RD$ trained baseline. In line with our main findings in Fig. 2c, full-model finetuning shows the biggest improvements for the high bitrate setting.

Interestingly, when comparing the low and high bitrate regimes, the total relative $RD\overline{M}$ gain of full-model finetuning follows a similar pattern for varying values of $f$ (higher gain for higher $f$). However, the subdivision of this gain in rate and distortion gain differs due to leveraging another tradeoff setting $\beta$. For the high bitrate, mainly distortion is diminished (row 2), whereas for the low bitrate, rate is predominantly reduced (row 3). These rate and distortion reduction plots clearly show how the flexibility of full-model (compared to encoder-only) finetuning can improve results in various conditions.

This experiment has shown that the potential for full-model finetuning (under the current model architecture and prior) seems highest for video compression purposes, as gains are negative (due to relative high static initial costs in the low bitrate regime) or only marginal (in the high bitrate regime) when overfitting on a single frame. Yet, we hypothesize that full-model finetuning could still be useful for (single) image compression as well, given other choices for the model architecture and/or model prior. Also, the provided ablation is run on one video only, so further research is needed to investigate full-model finetuning in an image compression setup.

# E   RATE-DISTORTION FINETUNING PERFORMANCE PER VIDEO

Figure 8 shows the $RD\overline{M}$ plots for the different videos after finetuning for 100k steps. Full-model finetuning outperforms the global model, *encoder-only* and direct latent optimization for all videos. The blue lines, indicating global model performance, differ per video, which might influence the finetuning gains, which also differ per video, e.g. controlled_burn versus sunflower. True entropy-coded results are used to create these graphs, rather than the computed $RD\overline{M}$ values. Deviations between entropy-coded and computed rates were found to be negligible (mean deviation was 1.94e-04 bits/pixel for $R$ and 1.06e-03 bits/pixel for $\overline{M}$). Throughout this paper, all training graphs and ablations are therefore provided using the computed values, rather than the entropy coded results.

Figure 9 shows for each of these videos the finetuning progression over training steps. Also here, differences in performance are visible among videos. The videos that result in highest finetuning gains, e.g. sunflower, show quicker performance improvement after the start of finetuning, and also continue to improve more over time.

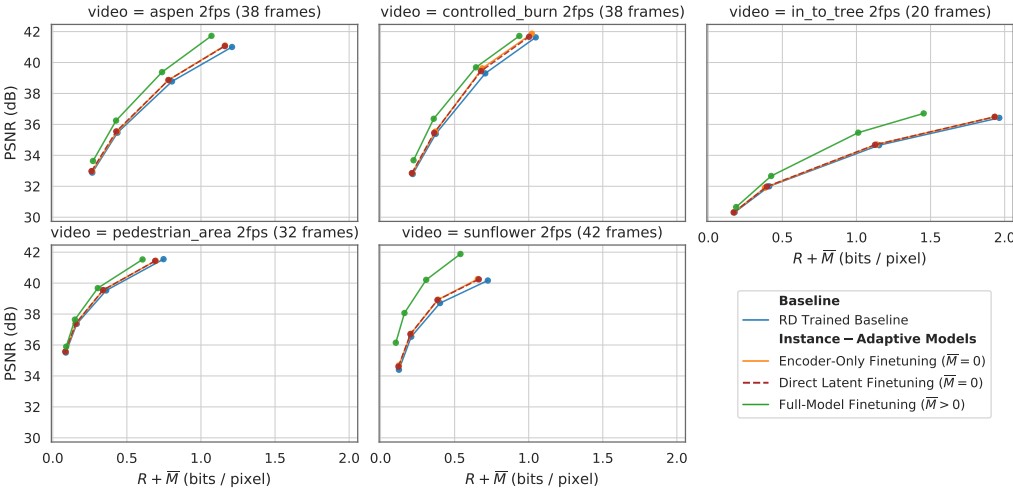

Figure 8: $RD\overline{M}$ performance for instance-adaptive encoder-only and full-model finetuning, compared with the performance of the global model, split per video. The instance-adaptive models used to create each graph are finetuned on the corresponding single video.

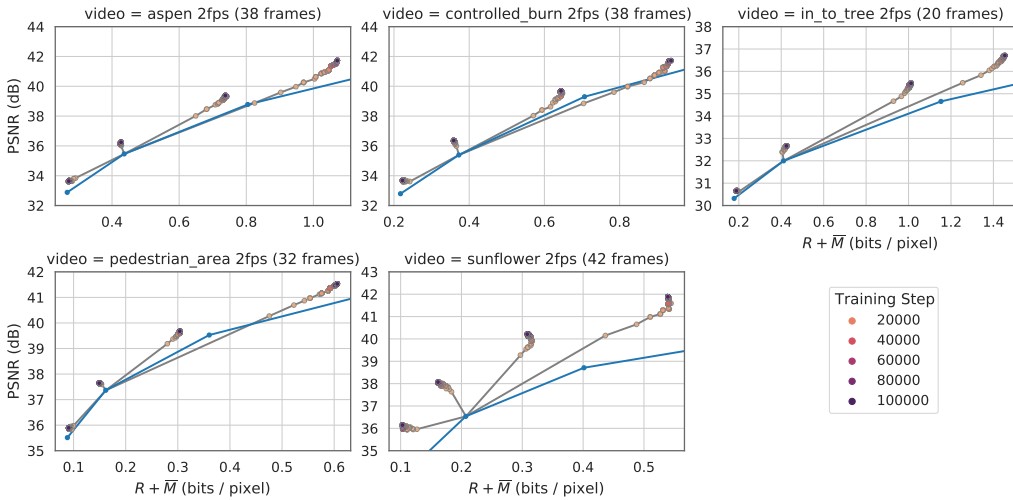

Figure 9: Progression of finetuning over time for all videos in Xiph-5N 2fps.

## F    MODEL UPDATES DISTRIBUTIONS

Figure 10 shows how the (quantized) model updates become much sparser (top row) when finetuning includes the spike-and-slab model rate loss $M$, compared to unregularized finetuning (bottom row).

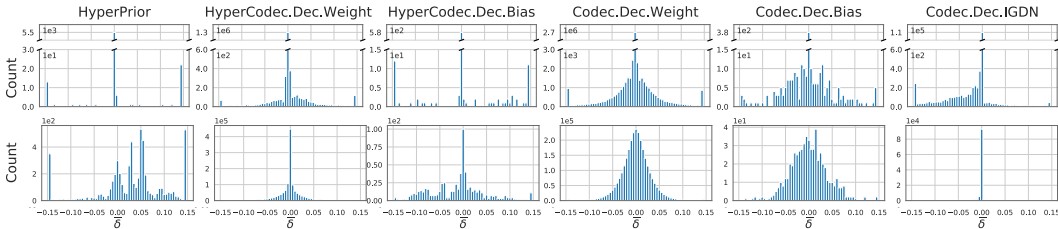

Figure 10: Histograms showing the distribution of quantized model updates $\bar{\bar{\delta}}$ when finetuning with (top row) and without (bottom row) the model rate regularizer $M$.

