# OpenReview forum: "Overfitting for Fun and Profit: Instance-Adaptive Data Compression"
_ICLR.cc/2021/Conference — ICLR 2021 Poster_

### Official Review · AnonReviewer2 · 2020-10-24
**Misleading results [updated]**

**Rating:** 6
**Confidence:** 4

**Review:**

Summary
-------------
This paper extends neural compression approaches by fine-tuning the decoder on individual instances and including (an update to) the decoder in the bit-stream for each image/video. The proposed approach is evaluated on the UVG dataset and the authors find a 1db improvement (PSNR) relative to their own baseline.


Quality (5/10)
----------
The proposed approach is sound and it would have been interesting to see the gains which can be achieved by fine-tuning the decoder of common neural compression approaches. Unfortunately, the few results provided in the paper are not just failing to answer this question but are misleading. By choosing a weak baseline, the reader is led to believe that fine-tuning a decoder will lead to large gains when realistic models are likely to benefit significantly less.

The authors motivate their simple baseline by noting that their approach is "model-agnostic". However, while the approach is model-agnostic, the results and conclusions are not. And it is mostly the empirical results which will be of interest to the reader. (A reader familiar with compression will be very well aware that a neural decoder _could_ be included in the bit-stream, making the conceptual contributions less interesting.)

The evaluated model encodes each frame of a 600 frame video sequence _independently_. A more realistic decoder would be conditioned on information in previously encoded frames, changing its behavior. It is reasonable to expect that similar change in behavior is encoded in the model updates. That is, the proposed approach is likely less effective in a more realistic setting.

If model complexity was a concern, the authors could have evaluated their approach on images instead of videos. The results would have looked less impressive but would have been more useful. Alternatively, they could have chosen a different video compression architecture of low complexity but one which is still practically relevant. E.g., one motivated by computational constraints.


Significance (4/10)
----------------
Neural compression is of interest to many people in the the ICLR community and exploring the fine-tuning of decoders would be a useful contribution to this field. The significance of this contribution is only limited by the lack of a meaningful results.


Originality (4/10)
--------------
Including model information in the bit-stream is an old idea in compression and not limited to neural compression. For example, Netflix is optimizing their classical video codecs at a "shot" level. Even JPEG (1992) allows us to fine-tune the Huffman table for an individual image ("optimized JPEG").

It is also common for compression challenges to require the model to be included in the bit-stream (e.g., the Hutter prize or the P-frame challenge of CLIC 2020).

Many papers have been written on the related topic of _model compression_ (e.g., Han et al., 2016), which should at least be acknowledged. Compressed model updates are also used in parallelized implementations of SGD (e.g., Alistarh et al., 2017).


Clarity (8/10)
---------
The paper is well written and clear.

---

> ### Comment · AnonReviewer2 · 2020-11-18
> **Updated experiments more interesting**
>
> The updated 2fps I-frame experiment is more realistic and thus provides more meaningful results. I still think the experimental section could have been stronger. For example, why not include results on single images? Even if they are negative, they would have increased the paper’s value to the community. Nevertheless, I increased my score from 4 to 6.

---

> > ### Author Response · Authors · 2020-11-18
> > **Thanks for reconsidering**
> >
> > We thank the reviewer for reconsidering his/her review and posting a reply again. We will answer the remaining concerns below:
> >
> > >  ### Quality
> > > ...
> >
> > We agree with the referee that our initial adoption of the I-frame model (i.e. all frames were I-frames) comprised a weak baseline. However, as already acknowledged by the reviewer, we now changed to a realistic setup, in which we sampled the I-frames at 2 frames per second as is common in actual video compression systems.
> >
> >
> > > "A reader familiar with compression will be very well aware that a neural decoder could be included in the bit-stream, making the conceptual contributions less interesting."
> >
> > The reviewer mentions that "a reader familiar with compression will be very well aware that a neural decoder could be included in the bit-stream". We indeed agree that familiar readers will be aware of the fact that inclusion of an updated model in the bitstream is a possibility. However, doing this while not greatly increasing the resulting rate is highly non-trivial, as can be seen by the fact that related work often focuses on encoder-only (Aytekin et al. (2018) & Lu et al. (2020)) or latent-only finetuning (Campos et al. (2019), Yang et al. (2020), Gou et al. (2020)), or only finetuning a (small) part of the decoding model (Lam et al., 2019;2020, Klopp et al. (2020)), rather than the full model. To the best of our knowledge, finetuning an entire neural network model (and showing larger RD gains), has never been done before.
> >
> > > "If model complexity was a concern, the authors could have evaluated their approach on images instead of videos. The results would have looked less impressive but would have been more useful."
> >
> > This original remark is close to the newly made request to report negative results on the use case where each instance is exactly one image/frame. We agree that reporting such negative results is in fact of interest to the reader of this paper. We aim to update the paper with such results. Depending on the computational resources we will have available the coming days, we hope to finish these experiments before the end of the rebuttal period.
> >
> > > "Alternatively, they could have chosen a different video compression architecture of low complexity but one which is still practically relevant. E.g., one motivated by computational constraints."
> >
> > To respond to the raised concern, we'd like to refer to [our earlier answer in the general thread](https://openreview.net/forum?id=oFp8Mx_V5FL&noteId=RTv80aNTt9E), explaining why moving to a lower-complexity video compression model might not be so trivial.
> >
> > > ### Significance (4/10)
> > > ...
> >
> > We hope that the previous answers have clarified that we mainly foresee big opportunities for full-model finetuning for video compression. The reported results show how our full-model finetuning framework has merit to greatly improve the sub-problem of I-frame compression in video compression.
> >
> > > ### Originality (4/10)
> > > Including model information in the bit-stream is an old idea in compression and not limited to neural compression. For example, Netflix is optimizing their classical video codecs at a "shot" level. Even JPEG (1992) allows us to fine-tune the Huffman table for an individual image ("optimized JPEG").
> >
> > We agree with the referee that the idea of including model information to the bit stream is not novel, and we did not intent to claim this. We already acknowledged related work that also finetuned (parts) of the decoding model (Lam et al., 2019;2020, Klopp et al., 2020). Yet, extending the bit stream with information regarding *full-model* updates is novel, and has to the best of our knowledge never been done before.
> >
> > > It is also common for compression challenges to require the model to be included in the bit-stream (e.g., the Hutter prize or the P-frame challenge of CLIC 2020).
> >
> > We acknowledge the reviewer's remark. However the concept is not equivalent. The Hutter prize deals with language models, which are conceptually different from video/image models, and in the P-frame challenge of CLIC2020 the entire model size is added to the compressed data size. The goal of such a model-size penalty is to promote small model designs over ever-growing (heavily over-parameterized) neural networks. In our use case we do not necessarily want to limit the total size of the model, yet we are interested in restricting the model updates under a given model prior used for entropy coding these updates.
> >
> >
> > > Many papers have been written on the related topic of model compression (e.g., Han et al., 2016), which should at least be acknowledged. Compressed model updates are also used in parallelized implementations of SGD (e.g., Alistarh et al., 2017).
> >
> > We agree with the referee that we lacked references to works in the model compression literature. As such, we have extended our related work section with a paragraph dedicated to model compression research, acknowledging (among others) the proposed references by the reviewer.

---

> > > ### Author Response · Authors · 2020-11-20
> > > **Added Temporal Ablation**
> > >
> > > We thank the reviewer for the comment regarding publishing negative results. We slightly extended this idea by doing an ablation experiment in which we varied the number of I-frames on which we finetuned ([see our reply in the general thread](https://openreview.net/forum?id=oFp8Mx_V5FL&noteId=AaOaYui8hV)).

---

### Official Review · AnonReviewer3 · 2020-10-28
**Good idea and sound method, but experiments can be better executed.**

**Rating:** 7
**Confidence:** 4

**Review:**

This paper considers the problem of per-instance model adaptation for neural data compression, and proposes a new method for end-to-end finetuning the model that is quantization-aware, by introducing an additional term that measures the compression cost of model update to the typical rate-distortion loss. Evaluation on the UVG dataset shows encouraging performance, with an average distortion improvement of approximately 1 dB for the same bit rate compared to the naive baseline (without fine-tuning).

------------------------

Pros:
1. The paper is well written and concepts are clearly explained.
2. The method is sound, and incorporating the entropy cost of model update during fine-tuning offers a conceptually appealing (and likely more performant, though not empirical verified (see below)) approach compared to previous methods (Lam et al., 2020, Zou et al., 2020) that tackles model update quantization after fine-tuning.

------------------------

Cons:
The experiment section is the weakest point. Particularly:
1. It's unclear from the description if the evaluation on UVG actually "adapts the entire model to a single data instance" (i.e., *for each image*) as claimed, or amortizes the model update cost over a batch of all the images in a video. The paper claims that "In this paper we consider the extreme case where the domain of adaptation is a single instance, resulting in costs for sending model updates which become very relevant", but this would highly misleading if all the experiments were conducted in a batch compression setting.
2. If the experiment did perform per-instance model adaptation, then it would be much more convincing to evaluate on standard datasets like Kodak and Tecnick from the image compression literature, instead of frames of UVG videos.
3. Since the paper's contribution is about improving the existing fine-tuning strategy that tackles model update quantization after fine-tuning (e.g., Zou et al., 2020), the proposed method should then also compare to these baselines to really assess its performance.
4. It would also be interesting to compare with approaches that optimize the encoded latents (e.g., Yang et al., 2020), which also achieve close to 1 PSNR improvement at equal bitrate without the overhead of decoder updates.

------------------------

Questions:
1. Can the author comment on how "the quantization bin width t and standard deviation σ of p[\bar δ]" (Sec 4.3) are chosen? How sensitive is the compression performance to their choice, e.g., is it possible to discretize so finely that no amount of RD improvement can overcome the model update cost?
2. The use of the continuous density for the M (model update cost) term in Eq 2 is established in the Appendix A by showing that the gradient of the discrete cost \bar M has the same gradient (up to first order) as that of -log p(δ) based on the density p(δ). Did I understand this correctly?  But M = -log p(δ) doesn't actually give an estimate of the cost after discretization \bar M = -log p[\bar δ]. Instead, the typical thing to do in literature (due to Balle et al.) is to actually minimize -log p[\bar δ], where \bar δ = round(δ), and the rounding can be either approximated by uniform noise injection or STE.   Can the authors comment on this choice of their method?

------------------------

Typos and minor mistakes/fixes:
1. p. 2, under eq (1): The R-D loss is equivalent to the *negative* ELBO in VAEs;
2. Does Figure 3 bottom show the histogram of bit allocation for \bar δ? If so then the caption can just say "Bottom: histogram of bit allocation for \bar δ" as it's clearer.

------------------------
Update after author response:

I have increased my score in light of the substantial improvement to the manuscript and experiments.

---

> ### Author Response · Authors · 2020-11-18
> **Response to AnonReviewer3 (1/2)**
>
> We thank the reviewer for his/her time to review our work. The raised concerns are answered below:
>
> > "It's unclear from the description if the evaluation on UVG actually "adapts the entire model to a single data instance" (i.e., for each image) as claimed, or amortizes the model update cost over a batch of all the images in a video. The paper claims that "In this paper we consider the extreme case where the domain of adaptation is a single instance, resulting in costs for sending model updates which become very relevant", but this would highly misleading if all the experiments were conducted in a batch compression setting."
>
> We agree with the referee that our initial formulation facilitated mis-interpretation. As explained in our reply in the general thread, we now changed to a realistic I-frame setup in which the model is adapted to a set of I-frames from one video (and amortize of all these frames). We rephrased the formulation in the paper to be more clear on the definition of one instance in our setup, and thereby hope to have taken away the reviewer's concern.
>
>
> > "Since the paper's contribution is about improving the existing fine-tuning strategy that tackles model update quantization after fine-tuning (e.g., Zou et al., 2020), the proposed method should then also compare to these baselines to really assess its performance."
>
> We agree with the reviewer that the original manuscript lacked evidence for the proposition that quantization-aware finetuning improves compression performance. We therefore added an ablation study in which we show that both quantization- (and model rate-) aware finetuning greatly improves performance.
>
>
> > "It would also be interesting to compare with approaches that optimize the encoded latents (e.g., Yang et al., 2020), which also achieve close to 1 PSNR improvement at equal bitrate without the overhead of decoder updates."
>
> We also agree with the referee that a baseline was missing in which the latents are optimized directly (as in Campos et al, 2019 & Yang et al., 2020). As such, we updated our experimental section with comparison to latent-only finetuning, which is shown to perform similarly to encoder-only finetuning.
>
> The additional framework-agnostic improvements proposed by Yang et al. (2020) (e.g. bits-back coding) in order to achieve a final gain of 1 dB, can in future research be added to our novel concept of full-model finetuning as well. In order to make a clean and fair comparison, we thus compare to latent-finetuning only without the additional improvements proposed.
>
> > ### Questions:
> >
> > Can the author comment on how "the quantization bin width t and standard deviation σ of $p[\bar{ \delta}]$" (Sec 4.3) are chosen? How sensitive is the compression performance to their choice,
>
> Both the quantization bin width $t$ and standard deviation $\sigma$ were empirically chosen, without major tuning. We initially run a naive, unregularized finetuning experiment to see in which order of magnitude the parameter updates would be distributed. Setting sigma=0.05 seemed to be an appropriate choice, which we did not tune further ever since. Thereafter we heuristically set the quantization bin width a factor 10 lower to 0.005. We additionally tested with a quantization bin width of 0.01 and found low sensitivity to this change in value.
>
> > is it possible to discretize so finely that no amount of RD improvement can overcome the model update cost?
>
> Indeed quantization can be so finely that the number of bits needed to encode each quantized update is so large that the resulting model update costs can not be overcome by an increase in RD performance. In this situation, when optimizing the RDM loss, no parameters will be finetuned (e.g. $\delta$ will remain $\mathbf{0}$) . As a consequence, the finetuned model will have identical distortion but $\bar{M}_0$ added to the rate. Thanks to our currently employed spike-and-slab prior, these static costs are rather small, and looking at the plots that indicate the finetuning progression over time (Appendix D, Fig. 8), we can also see that a net RD gain is being achieved directly at the start of finetuning.

---

> > ### Author Response · Authors · 2020-11-18
> > **Response to AnonReviewer3 (2/2)**
> >
> > > "The use of the continuous density for the M (model update cost) term in Eq 2 is established in Appendix A by showing that the gradient of the discrete cost \bar M has the same gradient (up to first order) as that of -log p(δ) based on the density p(δ). Did I understand this correctly? But M = -log p(δ) doesn't actually give an estimate of the cost after discretization \bar M = -log p[\bar δ]. Instead, the typical thing to do in literature (due to Balle et al.) is to actually minimize -log p[\bar δ], where \bar δ = round(δ), and the rounding can be either approximated by uniform noise injection or STE. Can the authors comment on this choice of their method? "
> >
> > We thank the referee for this interesting question. We confirm his/her understanding of Appendix A. Indeed the gradient of the continuous model rate penalty is (up to first order) equivalent to the gradient of its discrete counterpart, and indeed, the continuous penalty M does not give an estimate for the number of bits to be paid for the model update costs. This mismatch is caused due to a bias present between the number of bits and its continuous measure. Though, realize that a bias does not affect optimization behavior as it leaves the gradient unaffected, and thus gradient-based optimization as well. The fact that the continuous model rate costs itself are thus not a proxy for the actual number of bits to be paid does not matter while finetuning the model, as only the gradient is important to be a valid proxy.
> >
> > The referee proposes the use of the discrete model rate costs during training, including the Straight-through estimator to enable gradient updates. We indeed agree that the discrete bit rate overhead (as presented in App. A2, Fig. 4 (bottom)) could have been used for finetuning, as we indeed applied the Straight-through estimator to compute this gradient.
> > However, empirically we found the influence of finetuning with either the discrete or continuous model rate penalty negligible and therefore chose to adopt the continuous penalty as it might prevent unstable gradients that constantly switch (when being on the boundary between two quantization bins) during finetuning. Additionally, after updating the manuscript to use the proposed spike-and-slab prior ([see our reply from 13 November](https://openreview.net/forum?id=oFp8Mx_V5FL&noteId=Ap52XlEYB0l)), we extended Appendix A2 with a figure (orange in Fig. 4) showing the effect of the spike on the gradient. From that figure we see how the spike's effect on the gradient is almost fully canceled out due to quantization. This provides an extra reason why we finetune with the continuous regularizer.
> >
> > > "Typos and minor mistakes/fixes:"
> >
> > p. 2, under eq (1): The R-D loss is equivalent to the negative ELBO in VAEs;
> > We thank the reviewer for this remark and changed it in the updated manuscript.
> >
> > > Does Figure 3 bottom show the histogram of bit allocation for \bar δ? If so then the caption can just say "Bottom: histogram of bit allocation for \bar δ" as it's clearer.
> >
> > Indeed the bottom row in Fig. 3 shows how much bits are being paid per update level $\bar{\delta}$. We followed the referee's advice and changed the caption of this figure.
> >
> > ### References
> > - Joaquim Campos, Simon Meierhans, Abdelaziz Djelouah, and Christopher Schroers. Content adap- tive optimization for neural image compression. In Proceedings of the IEEE Conference on Com- puter Vision and Pattern Recognition Workshops, pp. 0–0, 2019.
> > - Yibo Yang, Robert Bamler, and Stephan Mandt. Improving inference for neural image compression.

---

> > > ### Comment · AnonReviewer3 · 2020-11-25
> > > **Thanks for the thorough response.**
> > >
> > > Thank you for responding to all my questions/concerns and improving the submission.
> > > I have increased my score in light of the substantial improvement to the method and experiments.
> > >
> > > A few nitpicks for the updated manuscript:
> > > 1. there was never a specification of what "quantization-aware training" is, so it's not clear how the ablations (II, III) actually remove the "quantization-aware training" component of the method;
> > > 2. there's a broken reference "discussion in Section ??." at the end of section 1.

---

> > > > ### Author Response · Authors · 2020-11-25
> > > > **Thanks for Reconsidering**
> > > >
> > > > We thank the reviewer for reconsidering his/her review and giving additional feedback. We will update the paper (in case of acceptance) accordingly.

---

### Official Review · AnonReviewer1 · 2020-10-28
**A nice idea, well communicated**

**Rating:** 7
**Confidence:** 4

**Review:**

This paper investigates how to improve the test time performance of learned image compression models through finetuning of the full model. The authors finetune the model (both the model parameters and the prior on the latent space) for every test-time instance, appending the model updates to the bitstream. The model updates are coded according to a discretised, mean-zero Gaussian distribution with a single learned variance. They demonstrate that this approach yields a superior rate-distortion curve than the non-finetuned model on a set of I-frame video data.

Overall I like the paper. It is clearly and simply written, with good motivation given for the concepts introduced. The method itself is also straightforward to understand, and seems like a sensible approach. Although on the surface the idea of doing instance-specific fine-tuning might seem to be impractical, it benefits from the fact that the extra encoding time of fine-tuning the model is paid by the sender. The receiver only has to pay the extra cost of decoding the model updates, which is fast if the coding distribution is factored (as it is in this paper). These asymmetrical coding times are often acceptable, as the authors note, since encoding-decoding is often a one-to-many relation.

I think the results demonstrated by the method are positive enough to warrant the extra overhead introduced, with a ~1dB gain for a given bitrate. I also appreciate the breakdown of where the extra model delta bits are allocated as per Figure 3, and the visualisation of the training performance in Figure 2b. I think these give a nice feel for the way the method works and the finetuning progresses on this particular instance.

Do the authors have any comment on why the encoding-only finetuning yields barely any benefit, as shown in Figure 2a? My interpretation might be that finetuning only the encoder is sub-optimal because the latent prior is fixed. The prior will have been learned jointly with the encoder on the global model, such that the encoder maps to parts of the latent space that the prior assigns mass to. As such, if the prior is fixed and you then finetune the encoder, the encoder still has to map to parts of space that are assigned mass in order to avoid the rate becoming too large. It might be interesting to see the results if the encoder and prior are finetuned but not the decoder. Although if you are finetuning (and communicating side information for the prior updates) then it is probably very little extra cost to also update the decoder. The results also seem to indicate that most bits for the model updates are spent on the decoder weights.

I also think it would have been good to include results using a learned prior to code the model updates, not a Gaussian. The authors do mention this as a possibility in the discussion, but surely it would have been very easy to implement? Given that they are already doing so for the latent space itself. Another small point about the Gaussian quantisation, is that an alternative discretisation is that of assigning equal mass to all bins, as per https://arxiv.org/abs/1901.04866 (see Appendix B). This results in simple coding - the discrete distribution is uniform, since the bins all have equal mass - and ensures that the discretisation is appropriate for the Gaussian.

---

> ### Author Response · Authors · 2020-11-18
> **Response to AnonReviewer1**
>
> We thank the reviewer for this positive and constructive review. We are pleased to read that the reviewer appreciates Fig. 2b and 3 specifically, as we indeed added those to provide the reader with insights behind the final results.
>
> ###  Remarks on encoder-only performance
>
> We agree with the interesting observation the referee makes regarding encoder-only finetuning. When we started this research, we initially investigated how (naive, unregularized) finetuning of different subsets of the model (e.g. encoder+prior or encoder+decoder) affected performance. We quickly noticed that best results were found when both (a subset of) prior and decoder were finetuned. Finetuning (part of) the prior (on top of encoder finetuning) faciliated reduction in rate, while finetuning (part) of the decoder reduced distortion. In order to achieve best RD performance, we thus concluded that full-model finetuning is definitely desirable. And indeed, we share the referee's hypothesis that only finetuning the encoder parameters (or latents directly) is limited by the fact that the latent prior is frozen.
>
> ### Learned model prior
>
> We find it interesting to read that the reviewer is as curious as we are to see how performance will benefit from using a learned prior, rather than a fixed Gaussian. A natural extension would indeed be to jointly train the standard deviation $\sigma$ and/or quantization bin width $t$ per parameter, while still restricting ourselves to Gaussian priors. We however foresee a situation where both $\sigma$ and $t$ collapse to extremely small values, resulting in a prior with (almost) zero-entropy. This would result in the initial costs $\bar{M}_0$ being zero and is therefore a trivial solution in which the model could easily collapse, making this natural extension possibly less trivial than expected.
> When moving to more complex and highly parameterized learned model priors, the question arises whether its parameters are fitted to a data instance or to a dataset of instances. In the first case, we need to signal its parameters in the bitstream which would likely be costly. When the prior is fitted over a dataset of instances, training might be expensive and there are no guarantees that the prior would generalize to unseen instances.
>
> The previous reasoning made us belief that the use of learned priors for our model prior can best be investigated in a separate, future research. Besides, we belief that we present an elegant and simple concept that already provides considerable gains. The fact that this framework already works using such a naive model prior, opens up a whole new field for future research in neural data compression.
>
> ### Quantization
>
> The suggestion of the reviewer to use quantization bins of equal mass rather than bins with uniform spacing is indeed interesting. It would increase the support for large model updates, possibly without (a large) increase in model rate, and it faciliates finer quantization for the small updates. As we also wanted to improve upon the relatively large initial static costs in our revised manuscript, we upgraded our model prior to a spike-and-slab prior (see [our response of 13 Nov](https://openreview.net/forum?id=oFp8Mx_V5FL&noteId=Ap52XlEYB0l)). To not induce multiple changes at the same time, we leave the extension to use equal-mass quantization bins for future research.

---

> > ### Author Response · Authors · 2020-11-20
> > **Added Temporal Ablation Experiment**
> >
> > The initial review of the referee asked whether we could explain the relatively low performance of encoder-only finetuning. As was already suggested by the referee back then (and acknowledged by us in the reply), the encoder/latents are more difficult to be finetuned when the prior and decoder model are frozen. This hypothesis is confirmed by [our new ablation experiment](https://openreview.net/forum?id=oFp8Mx_V5FL&noteId=AaOaYui8hV) in Appendix D (Fig. 7), which shows that full-model finetuning is able optimize rate (for low bit rate regime), and distortion (for high bit rate regime) much more than encoder/latent-only finetuning.

---

### Official Review · AnonReviewer4 · 2020-10-28
**Instance specific finetuning method for image and video compression but with weaknesses in the experimental section**

**Rating:** 6
**Confidence:** 3

**Review:**

**Summary**

The paper describes an instance specific finetuning method for image and video compression including finetuning the decoder. Based on the shown experiments, the required additional bits for sending the updated finetuned model parameters are worth the achieved increase in RD performance. However, the method has only been evaluated on one video dataset and with respect to its own baseline and not with respect to any other existing method.

**Strength**

= Method which also considers to finetune/adapt the decoder side of image compression network, for improved performance.

= Paper is self-contained by recapping the necessary basic formulations.

**Weakness**

= Method has only been evaluated with respect to its own baseline method (image compression model without finetuning).

= Method has only been evaluated on one video dataset, but by compressing frame by frame, therefore not taking advantage of temporal redundancy.

= Given that it is an image compression method, the proposed instance adaptive method could also be evaluated on the e.g. clic validation set.

*Some open questions*

Is $\bar{M}$ computed for the whole video and averaged per frame for the results in Table 1 and therefore dependent on the length of the video?

Do the authors have some intuition, why some videos are easier to finetune than others?

*Minor*

References of arxiv papers, which have been published before submission deadline, can be updated with the respective conference.

---

> ### Author Response · Authors · 2020-11-18
> **Response AnonReviewer4**
>
> > "Method has only been evaluated with respect to its own baseline method (image compression model without finetuning)".
>
> We follow the reviewer's advice, and now implemented direct latent optimization as proposed by Campos et al. (2019), and later used as well by Yang et al. (2020) and Guo et al. (2020), next to the already present encoder-only finetuning baseline. Besides, the non-finetuned baseline model is the de-facto neural image compression standard nowadays (see [our general reply from 13 November](https://openreview.net/forum?id=oFp8Mx_V5FL&noteId=Ap52XlEYB0l)), making it another valid baseline to show the merit of this new concept in our opinion.
>
> > "Method has only been evaluated on one video dataset, but by compressing frame by frame, therefore not taking advantage of temporal redundancy."
>
> We indeed show results of our framework on one dataset, but one should realize that in typical machine learning setups, one dataset entails one training where the model learns to capture the statistics of this dataset. In our case, for each video in this dataset a new model is being finetuned, making each video an experiment on its own, as each video's characteristics differ. We've changed to the Xiph dataset (see [our reply from 13 November](https://openreview.net/forum?id=oFp8Mx_V5FL&noteId=Ap52XlEYB0l)), and the selected videos vary in many aspects including framerate, camera used for shooting, single-shot vs multi-shot and clip content.
>
>
> The remark regarding ignoring temporal redundancy has been raised by multiple reviewers. In response, we now changed our all-intra frames setup (i.e. all frames are I-frames), to a realistic use case of I-frame compression at 2 fps. For more details we refer again to [our general reply from 13 November](https://openreview.net/forum?id=oFp8Mx_V5FL&noteId=Ap52XlEYB0l).
>
>
> > "Given that it is an image compression method, the proposed instance adaptive method could also be evaluated on the e.g. clic validation set."
>
> We thank the reviewer for the suggestion. As AnonReviewer3 made the same remark as an answer to our general reply, our answer [can be found there](https://openreview.net/forum?id=oFp8Mx_V5FL&noteId=RTv80aNTt9E).
>
> Some open questions
>
> > "Is $\bar{M}$ computed for the whole video and averaged per frame for the results in Table 1 and therefore dependent on the length of the video?"
>
> The referee indeed understood correctly that $\bar{M}$ was (in the original version of the paper) computed over the whole video, as finetuning also took place on the entire video. As we now overfit the full model on I-frames from videos that are only sampled at 2 fps (see our reply in the general thread), the model rate costs are also amortized over only these frames. We belief we made this more clear in the updated version of the manuscript. In Table 1 we initially provided the costs of $\bar{M}$ both in bits/pixel and bits/parameter. The former thus averages the costs over the pixels of all I-frames, making it dependent on the number of frames. The latter is dependent on the number of trainable parameters in the model and thus depends on the chosen model architecture. Upon this remark of the reviewer, we realized that it might also be of interest to the reader to see model rate expressed in bits or bytes per frame. As such, we extended Table 1 with an expression of M in this unit as well.
>
>
> > "Do the authors have some intuition, why some videos are easier to finetune than others?"
>
> We thank the reviewer for this interesting question, indeed finetuning gain differs among videos. Note that the performance of the global model for each video differs already, therewith influencing the maximum gains to be achieved by finetuning. Also, video characteristics such as motion and frequency content greatly influence the diversity of the set of I-frames, thereby affecting the ease of model-adaption.
>
> > "References of arxiv papers, which have been published before submission deadline, can be updated with the respective conference. "
>
> We thank the reviewer for this comment and updated all references with the appropriate conference or journal where possible.
>
> **References**
> - Joaquim Campos, Simon Meierhans, Abdelaziz Djelouah, and Christopher Schroers. Content adap- tive optimization for neural image compression. In Proceedings of the IEEE Conference on Com- puter Vision and Pattern Recognition Workshops, pp. 0–0, 2019.
> - Yibo Yang, Robert Bamler, and Stephan Mandt. Improving inference for neural image compression. Advances in Neural Information Processing Systems, 33, 2020b.
> - Tiansheng Guo, Jing Wang, Ze Cui, Yihui Feng, Yunying Ge, and Bo Bai. Variable rate image compression with content adaptive optimization. In Proceedings of the IEEE/CVF Conference on Computer Vision and Pattern Recognition Workshops, pp. 122–123, 2020.

---

### Author Response · Authors · 2020-11-13
**Manuscript Updates**

We thank all reviewers for the time to review our work and for the constructive feedback. Analyzing the remarks that were shared across reviewers, we decided upon improving the paper by four main updates, discussed below. Reviewer-specific remarks and our responses to those will be addressed below each review separately.

## Baseline model
Multiple reviewers found the presented baseline naive or unrealistic. Two reviewers address that temporal redundancy in videos is not taken into account in our model as we use an I-frame model where each image from a video is independently compressed. We acknowledge that the adopted I-frame frequency of 120 fps (i.e. all frames are I-frames) does not resemble a typical video compression setup, as in practice each second typically comprises only one or two I-frames. These I-frames are then still independently compressed to enable random access at any point in time. As such, we decided to change our setup to a more realistic setting where we independently compress I-frames sampled at 2 fps.

We would like to remark that the chosen mean-scale hyperprior model  (without autoregressive context-model) (Ballé et al., 2018; Minnen et al., 2018) is the de-facto standard for neural image compression (Yang et al., 2020; Agustsson & Theis, 2020; Chen & Ma, 2020) and is also commonly used in video compression works where I-frames are compressed using a neural network (Agustsson et al., 2020; Djelouah et al., 2019; Lu et al., 2019). We benchmarked our implementation of this model against the reported performance in the original paper and could reproduce their performance, providing support that our implementation of this standard for I-frame compression can be used as a valid and near state-of-the-art baseline model.

## Dataset
We noticed that some confusion was present about the amortization of the additional model update costs. In the original submission, we amortized the bit rate overhead over all frames of the video, as we were also finetuning the model on the entire stack of frames. For the UVG dataset this comprised 600 frames (5 seconds, sampled at 120 fps). However, as explained in the previous paragraph, the updated experiments will only compress two frames per second from each video. As the bit rate overhead is amortized over the resulting total number of I-frames, we decided to move to the [Xiph dataset](https://media.xiph.org/video/derf/) that contains longer videos (10-20 seconds). An additional benefit of Xiph over UVG is its increased variety of video characteristics.

## Model prior
Although optimization using our proposed $RDM$ loss automatically trades off the model update costs against the $RD$ improvements, the initial costs $\bar{M}_0$ are not part of this optimization. By switching to a realistic I-frame frequency, the total number of frames for amortization of the bit rate overhead is heavily reduced, and the static initial costs are rather large.
We therefore updated our model prior by increasing the probability mass for zero-updates ($\bar{\delta}=0$) by adding a narrow Gaussian around this zero-update. In the revised manuscript we show that this updated prior is a generalization of the earlier proposed Gaussian model prior, and that the resulting updates are sparser. Finetuning with this new prior works well for both short and long videos, making the proposed method more generally applicable.

## Benchmarking
We were asked to compare to methods that apply post-finetuning quantization (Lam et al., 2020, Zou et al., 2020), as one of our main contributions is quantization-aware finetuning. We agree that our initial manuscript lacked experimental evidence showing that quantization-aware finetuning indeed improves final compression performance. As such, we will update the manuscript with an ablation experiment to quantify this quantization gap. Our results show that this gap is substantial; supporting our claim that quantization-aware training improves compression performance.

### References
 - Eirikur Agustsson and Lucas Theis. Universally quantized neural compression.
 - Eirikur Agustsson, David Minnen, Nick Johnston, Johannes Balle, Sung Jin Hwang, and George Toderici.  Scale-space flow for end-to-end optimized video compression.
 - Johannes Ballé, David Minnen, Saurabh Singh, Sung Jin Hwang, and Nick Johnston.  Variational image compression with a scale hyperprior.
 - Tong Chen and  Zhan  Ma.   Variable bitrate  image compression with  quality  scaling factors.
 - Abdelaziz Djelouah, Joaquim Campos, Simone Schaub-Meyer, and Christopher Schroers.  Neural inter-frame compression for video coding.
 - Guo Lu, Wanli Ouyang, Dong Xu, Xiaoyun Zhang, Chunlei Cai, and Zhiyong Gao.  Dvc: An end-to-end deep video compression framework.
 -  David Minnen, Johannes Ballé, and George D Toderici.  Joint autoregressive and hierarchical priors for learned image  compression.
 - Yibo Yang, Robert Bamler, and Stephan Mandt. Improving inference for neural image compression.

---

> ### Comment · AnonReviewer3 · 2020-11-13
> **Not sure about doing evaluation on another video dataset...**
>
> Thanks for this detailed update and addressing some of our common concerns.
>
> I have one quick suggestion: as reviewer 2, reviewer 4, and I have already pointed out, since the model and evaluation setup are mainly targeted at (batch) image compression (and considered naive and impractical for video compression), it would be really helpful to also see (batch) image compression results on standard image datasets like Kodak, Tecnick, or CLIC validation set.
> Alternatively, if the focus is really on video compression, then working with say a lower-complexity (e.g., computationally constrained) yet still practically useful video compression model would also make the results more meaningful, as suggested by reviewer 4.

---

> > ### Author Response · Authors · 2020-11-18
> > **Our new I-frame compression setup is a realistic and common use-case in video compression.**
> >
> > We thank the reviewer for his/her quick reply and would like to take the opportunity here to clarify the scope of our experiments in relation to the real-world use case of I-frame compression in video compression. Typical video compression comprises of independent compression of key frames (I-frames), followed by conditional compression of the remaining frames.  For example Lu et al. (2019), Liu et al. (2020), Wu et al. (2018), Djelouah et al. (2019), and Yang et al. (2020a)  also all compress every 8th-12th I-frame independently. In this work we specifically tackle this I-frame compression subproblem of video compression. Since we show 1 dB compression gains on this task, the next step would be to finetune the P-frame (Lu et al. (2019), Liu et al. (2020), Yang et al. (2020a)) or B-frame (Wu et al. (2018), Djelouah et al. (2019)) model on the other frames by minimizing the $RDM$ loss amortized over those frames. We leave this as an exercise for future work.
> >
> > Even though we focus on the problem of I-frame compression, this is not the same as image compression. When applying our method for image compression, we would finetune a model for each image in a dataset and amortize the model rate $M$ over the number of pixels in that image. The high number of parameters per pixel would make it very difficult to reach good compression performance when taking into account the model rate.
> >
> > Instead, we want to amortize the cost of finetuning the model over multiple images or frames. In batch-image compression, a batch of various (uncorrelated) images is to be compressed jointly. This leads to a very specialized and uncommon use-case, as it requires knowledge regarding the exact set of images the user wants to be compressed.
> > The problem of video compression lends itself very naturally for full-model adaptation, as a user typically wants to receive the entire video.
> >
> > As we also indicated in our discussion section, we agree that leveraging low-complexity video models is an interesting application of our full-model finetuning framework. However, the necessary neural architecture search to find such a model (which has high enough capacity to adapt to a full video, but is at the same time small enough), results in a non-trivial problem which we leave for future research. On the contrary, we chose a state-of-the-art I-frame compression model to showcase our framework.
> >
> > As mentioned before, we now changed our all-intra frames setup (i.e. all frames are I-frames), to a realistic use case of I-frame compression at 2 fps. We have also updated our paper with more extensive explanations regarding our choice for the I-frame video compression use case, and therefore hope to take away concerns using the scope of this work.
> >
> >
> > **References**
> > - Yang Yang, Guillaume Sauti`ere, J Jon Ryu, and Taco S Cohen. Feedback recurrent autoencoder. InICASSP 2020-2020 IEEE International Conference on Acoustics, Speech and Signal Processing(ICASSP), pp. 3347–3351. IEEE, 2020a.
> > - Guo Lu, Wanli Ouyang, Dong Xu, Xiaoyun Zhang, Chunlei Cai, and Zhiyong Gao.  Dvc: An end-to-end deep video compression framework.   InThe IEEE Conference on Computer Vision andPattern Recognition (CVPR), June 2019.
> > - Haojie Liu, Han Shen, Lichao Huang, Ming Lu, Tong Chen, and Zhan Ma. Learned video compres-sion via joint spatial-temporal correlation exploration. InProceedings of the AAAI Conference onArtificial Intelligence, volume 34, pp. 11580–11587, 2020.
> > - Abdelaziz Djelouah, Joaquim Campos, Simone Schaub-Meyer, and Christopher Schroers.  Neuralinter-frame compression for video coding.  InProceedings of the IEEE International Conferenceon Computer Vision, pp. 6421–6429, 2019.
> > - Chao-Yuan Wu, Nayan Singhal, and Philipp Krahenbuhl.  Video compression through image inter-polation. InProceedings of the European Conference on Computer Vision (ECCV), pp. 416–431,2018.

---

> > > ### Author Response · Authors · 2020-11-20
> > > **Added Temporal Ablation**
> > >
> > > Some of the reviewers asked us to perform an image-compression experiment. Though we [explained why  full-model finetuning is non-beneficial for (single) image compression](https://openreview.net/forum?id=oFp8Mx_V5FL&noteId=RTv80aNTt9E), AnonReviewer 2 noted that it would still be valuable to include such (possibly negative) results in the paper.
> > >
> > > We definitely agree with publishing negative results, and therefore have now updated our manuscript with a temporal ablation, where for one video and two values of $\beta$ we repeat our main experiment for different numbers of frames sampled from the video.
> > >
> > > We show that full-model finetuning outperforms the encoder/latent-only finetuning methods, even for a really low number of frames. Full-model finetuning is found to be too costly when finetuning on 1 frame at the lowest bitrate. We believe that the added ablation study is of strong interest to the compression community as it clarifies the current boundaries of full-model finetuning.
> > >
> > > ~~_Note: As the experiments for the temporal ablation are not yet finished, we only used data from the first 30.000 finetuning steps to create Figure 7. This is the largest number of steps for the slowest run, ensuring a fair comparison at this moment. In case of acceptance we will update the figure with results after training for 100.000 steps (our default). Since most improvements are achieved in the early stages of training (see Figure 2b) we do not expect the results to change meaningfully._~~

---

### Author Response · Authors · 2020-11-25
**Revision Summary**

As today the rebuttal period ends, we would finally like to thank all reviewers for their time and constructive feedback, which greatly improved the quality of our work. To summarize; with respect to the initial submission, the following changes were made:

- We changed our experiments to a realistic setting where we finetune an I-frame compression model on a set of I-frames (sampled at 2 fps) for various videos, resulting in a considerable gain of (on average) 1 dB at the same bit rate.
- We updated our model prior to a spike-and-slab prior (p. 4, Section 3.2), such that the model itself can learn which parameters are worth the update and which aren't (and are therefore negligibly cheap to encode).
- We added two ablation experiments: one that separately quantifies the performance gains thanks to quantization- and model rate-aware finetuning, and one that investigates the influence of the number of I-frames on final performance (including the image-compression setup where we finetune on a single I-frame). The first ablation shows that both quantization- and model rate-aware finetuning greatly improve the compression performance (p. 8, Fig. 2c; p. 19 Fig 10), while the second ablation demonstrates that full-model finetuning with the spike-and-slab model prior works well for a wide range of number of frames (p. 17, Appendix D, Fig. 7).
- We added an extra baseline (p. 8, Fig. 2a); direct latent optimization (Campos et al., 2019), and we now report for four different beta values (rather than three in the initial submission).

We hope that these updates take away the reviewers' original concerns.

---

### Decision · Program_Chairs · 2021-01-07
**Final Decision**

**Decision:**

Accept (Poster)

**Comment:**

The paper suggests a procedure to efficiently adapting a learned neural compression model to a new test distribution. If this test distribution has low entropy (e.g., a video as a sequence of interrelated frames), large compression gains can be expected. To achieve these gains, the method adapts the decoder model to the new instance, transmitting not only the data but also a compressed model update. Experiments are carried out on compressing I-frames from videos, while comparisons comprise baseline approaches that finetune the latent representations of videos as opposed to the decoder.

The paper’s main contribution is very timely and relevant. While it was well-known in the classical compression literature that model updates could be sent along with the data (e.g., as already done in “optimized JPEG”), this is the first time the idea was implemented in neural compression. The experiments are arguably the paper’s weaker part and were originally a concern, but they have been significantly improved during the review period such that all reviewers voted for acceptance. We encourage the authors to further strengthen their experimental results by adding more challenging baselines on well-established tasks (e.g., image compression).